# Similar object shape representation encoded in the inferolateral occipitotemporal cortex of sighted and early blind people

Yangwen Xu[1]*, Lorenzo Vignali[1,2], Federica Sigismondi[1], Davide Crepaldi[2], Roberto Bottini[1], Olivier Collignon[1,3,4]*

1 Center for Mind/Brain Sciences (CIMeC), University of Trento, Trento, Italy, 2 International School for Advanced Studies (SISSA), Trieste, Italy, 3 Psychological Sciences Research Institute (IPSY) and Institute of NeuroScience (IoNS), University of Louvain, Louvain-la-Neuve, Belgium, 4 School of Health Sciences, HES-SO Valais-Wallis, The Sense Innovation and Research Center, Lausanne and Sion, Switzerland

* yangwen.xu@unitn.it (YX); olivier.collignon@uclouvain.be (OC)

**Data Availability Statement:** All relevant data are within the paper and its Supporting Information files.

## Abstract

We can sense an object's shape by vision or touch. Previous studies suggested that the inferolateral occipitotemporal cortex (ILOTC) implements supramodal shape representations as it responds more to seeing or touching objects than shapeless textures. However, such activation in the anterior portion of the ventral visual pathway could be due to the conceptual representation of an object or visual imagery triggered by touching an object. We addressed these possibilities by directly comparing shape and conceptual representations of objects in early blind (who lack visual experience/imagery) and sighted participants. We found that bilateral ILOTC in both groups showed stronger activation during a shape verification task than during a conceptual verification task made on the names of the same man-made objects. Moreover, the distributed activity in the ILOTC encoded shape similarity but not conceptual association among objects. Besides the ILOTC, we also found shape representation in both groups' bilateral ventral premotor cortices and intraparietal sulcus (IPS), a frontoparietal circuit relating to object grasping and haptic processing. In contrast, the conceptual verification task activated both groups' left perisylvian brain network relating to language processing and, interestingly, the cuneus in early blind participants only. The ILOTC had stronger functional connectivity to the frontoparietal circuit than to the left perisylvian network, forming a modular structure specialized in shape representation. Our results conclusively support that the ILOTC selectively implements shape representation independently of visual experience, and this unique functionality likely comes from its privileged connection to the frontoparietal haptic circuit.

## Introduction

Object properties can be accessed through multiple sensory channels. For example, knowledge of an object's shape can be acquired both by vision and touch. This brings up a critical question about the cerebral architecture of object representation: Are shape representations derived

**Funding:** This work was supported by the Belgian Excellence of Science (EOS) program (Project No. 30991544) attributed to O.C., the Research Projects of National Interest (PRIN) grant from the Italian Ministry of Education, University and Research (MIUR) attributed to D.C. and O.C. (Project No. 2015PCNJ5F_001), the Flag-ERA HBP PINT-MULTI (R.8008.19) attributed to O.C., and a mandate d'impulsion scientifique from the Fonds National de la Recherche Scientifique de Belgique (FRS-FNRS) attributed to O.C. O.C. is a research associate at the Fonds National de la Recherche Scientifique de Belgique (FRS-FNRS). The funders had no role in study design, data collection and analysis, decision to publish, or preparation of the manuscript.

**Competing interests:** The authors have declared that no competing interests exist.

**Abbreviations:** AG, angular gyrus; aIPS, anterior intraparietal sulcus; BA, Broadman area; EB, early blind; fMRI, functional magnetic resonance imaging; FOV, field of view; FWE, family-wise error; FWHM, full width at half maximum; GLM, general linear model; HRF, hemodynamic response function; IFG, inferior frontal gyrus; ILOTC, inferolateral occipitotemporal cortex; IPS, intraparietal sulcus; IS, independent sighted; LOC, lateral occipital complex; LOtv, lateral occipital tactile visual; pIPS, posterior intraparietal sulcus; RC, rotated component; RDM, representational dissimilarity matrix; ROI, region of interest; RSA, representational similarity analysis; RSFC, resting-state functional connectivity; RT, reaction time; SC, sighted control; SMG, supramarginal gyrus; STG, superior temporal gyrus; TMS, transcranial magnetic stimulation; vPMC, ventral part of the premotor cortex.

from different senses segregated from each other in the human brain, or, alternatively, could the brain implement a shared representation of object shape that is abstracted from the senses (e.g., [1–4])?

Cognitive neuroscientists usually investigate object shape representation along separate visual and haptic brain pathways. Studies on visual shape representation mostly focus on the ventral visual pathway in the occipitotemporal cortex. Researchers found that the lateral occipital cortex and the posterior fusiform gyrus (i.e., the lateral occipital complex, LOC) show greater activation to object images than texture images (see review [5]). By contrast, the medial part of the visual cortex is more sensitive to visual texture than visual shape (e.g., [6–8]). Lesions in the LOC induce visual form agnosia manifested as impaired shape discrimination but preserved texture discrimination performance [9,10], whereas lesions in the medial part of the visual cortex cause the opposite syndrome (e.g., [11]).

Studies on haptic shape representation highlighted the neural circuit in the ventral fronto-parietal cortex. Researchers found that the intraparietal sulcus (IPS; e.g., [12–15]) and the ventral part of the premotor cortex (vPMC, e.g., [14,15]) show greater activation when participants touch objects than textures. Lesions in the superior parietal cortex and the adjacent IPS induce contralateral tactile agnosia characterized by somatosensory discrimination deficits in the macrogeometrical domain (i.e., detecting differences in length of cuboids) but not in the microgeometrical domain (i.e., detecting subtle differences in grating profiles), whereas lesions in the postcentral gyrus cause the opposite syndrome [16]. Lesions in the anterior IPS (aIPS) and vPMC can also impair contralateral object exploration—patients cannot recognize objects haptically due to the disturbance of finely tuned finger movements, specifically when interacting with objects [17,18]. In the macaque brain, the homologous regions of both the aIPS (i.e., the AIP) and the vPMC (i.e., the F5) host the neurons that fire when monkeys configure their hands to grasp objects in particular shapes (e.g., [19,20]).

In addition to the frontoparietal circuit, haptic shape perception intriguingly involved the anterolateral part of the LOC, a region located in the inferolateral occipitotemporal cortex (ILOTC); this region shows stronger activation when participants both see or touch objects in comparison to shapeless textures (e.g., [12–15]). Based on this unique multisensory property, researchers termed the ILOTC region the lateral occipital tactile-visual complex (LOtv, [13]) and suggested it implements supramodal shape representation [3]. However, the nature of ILOTC remains debated, as current findings could also support alternative hypotheses.

First, the LOTC might engage in haptic tasks simply due to visual imagery. This hypothesis is supported by studies showing that experiences of visual imagery during haptic shape perception are common, and ratings of the vividness of visual imagery strongly predict the amount of haptic shape-selective activity in the right LOC [21]. To test whether visual imagery is a prerequisite for ILOTC's involvement during nonvisual tasks, two studies have tested early blind participants who lack visual imagery. These two studies, however, do not allow to settle the debate. One study found ILOTC's activation when contrasting a haptic object recognition task and a task imitating the grasping and exploration of objects [22]. Since this study did not match the two contrasted conditions on task demand and object semantics (see next paragraph), the isolated cognitive components might not be specific to shape processing. The other study, instead, did not find that the ILOTC-encoded object shape in the early blind participants and localized shape representation in other occipitotemporal regions [23]. Participants in this study performed a shape-irrelevant task (i.e., size judgment task), which might have dampened the brain activation relating to shape representation in the ILOTC.

Second, the ILOTC might engage in conceptual representation of objects. An object does not only have a shape, it carries meaning and serves a function. Whenever in the above contrasts between objects and textures (e.g., [12,13]) or between the haptic condition with objects

and the hand movement condition without objects [22], the isolated cognitive component could be conceptual, not perceptual. Previous studies have indeed suggested that the ventral visual pathway might encode semantic relatedness among objects (e.g., [24]). This is even more likely for the ILOTC. Regions overlapping or slightly superior to the ILOTC show category preference for manmade objects that persists in the early blind participants (e.g., [25–28]), and patients with lesions in the left lateral occipitotemporal cortex are slower to make conceptual associations among manmade objects (e.g., hammer-nail) [29]. However, this hypothesis was challenged by a recent study showing that the activity pattern in the ILOTC can encode object shapes when stimuli are meaningless novel shape models [30]. Nevertheless, these findings cannot rule out ILOTC's involvement in conceptual representation; the ILOTC might support an integrative coding of both visual and conceptual knowledge, as already shown in some other regions in the ventral visual stream [31].

Third, the ILOTC might engage in (visual) shape representation in the sighted but conceptual representation in the early blind. The pluripotent neuroplasticity hypothesis predicts that the "visual" cortex in the early blind, due to a lack of visual input since birth, could repurpose its function for cognitive faculties that are distant from its native computation in vision, like language or mathematics (see review [32]). This neurofunctional reorganization process usually accompanies enhanced connectivity between the "visual" cortex in the early blind and high-order brain systems [32]. In line with this hypothesis, it has been reported that the "visual" cortex in the early blind is more sensitive to lexical semantics than the sighted participants (e.g., [33,34]). Moreover, the activity in the lateral occipital cortex in the early blind is more synchronized to the areas in the perisylvian language network than in the sighted participants [34]. It is thus possible that the ILOTC in the early blind implements conceptual instead of shape representation due to functional reorganization.

To address these unsolved questions comprehensively in a single study, we used functional magnetic resonance imaging (fMRI) to characterize the brain activity of sighted and early blind participants when they were performing both shape and conceptual verification tasks on the same set of auditory words referring to manmade objects. Univariate contrast between shape and conceptual tasks was performed to localize brain areas specific for shape or conceptual processing. We chose words instead of haptic objects as stimuli because words are arbitrary symbols bearing no obvious resemblance to the objects signified. That means the words, per se, do not carry object information and can elicit shape and conceptual representations without bias. In contrast, haptic objects carry shape information. The participants would have to process the shape information to recognize the objects in both shape and conceptual tasks, and we could no longer isolate the shape representation by contrasting the shape task with the conceptual task. Besides task manipulation, we also orthogonalized the pairwise shape similarity and the pairwise conceptual association among the objects we selected (e.g., a "plate" is perceptually similar to a "coin" in shape but is conceptually associated with a "fork" in function). Representational similarity analyses (RSA, [35]), therefore, can be conducted to disentangle the regions implementing shape and conceptual representations. Furthermore, we used resting-state functional connectivity (RSFC) to detect the possible synchronizations between the ILOTC and the frontoparietal haptic network or the perisylvian language network.

If the ILOTC implements supramodal shape representation, we should find the ILOTC showing greater activation in the shape task than in the conceptual task in both sighted and early blind participants, and the activity pattern in the ILOTC should encode objects' shape but not conceptual properties. The ILOTC is expected to have stronger connections to the frontoparietal haptic network than the perisylvian language network. If the ILOTC represents objects' conceptual knowledge instead, we should observe greater activation in the conceptual task than in the shape task in both sighted and early blind participants, and the activity pattern

in the ILOTC should encode objects' conceptual properties. Alternatively, if the activation in the ILOTC depends on visual experience, we should observe the ILOTC's involvement in shape processing/representation only in the sighted but not in the early blind participants. If such "visual" ILOTC repurposes its function to conceptual representations in the early blind, we should find the ILOTC's involvement in conceptual representation only in the early blind but not in the sighted participants.

## Results

### Behavior rating on shape similarity and conceptual association

In this study, we selected 21 Italian words, which referred to 21 manmade objects, as our stimuli. The selection was mostly based on behavior ratings of object properties from an independent group of sighted participants who did not take part in the fMRI experiment ($N = 19$; see Stimuli in Materials and methods about the stimulus selection procedure and criteria). To validate the rating results from the stimulus selection stage and to verify whether the early blind population had a similar shape and conceptual knowledge as the sighted control, all participants who took part in the fMRI experiments ($N = 48$) also rated the object properties of the stimuli selected (Fig 1). These participants consisted of three groups: 16 early blind (EB) participants, 16 gender- and age-matched sighted control (SC) participants, and 16 independent sighted (IS) participants (see Participants in Materials and methods for details).

Shape similarity and conceptual association were rated on a 7-point Likert scale in a pairwise manner (see Procedures in Material and methods about the rating procedure). We assessed the inter-rater reliability within each group of participants using the intraclass correlation based on a mean-rating, consistency, two-way random model (i.e., ICC(C,k)) [36]. Both shape rating (ICC(C,k): 0.953–0.973) and conceptual rating (ICC(C,k): 0.984–0.985) showed "excellent" inter-rater reliability [37] (S3 Table). We averaged the rating scores within each group and compared them across groups. Fig 1A illustrates that the rating scores on both object properties were highly reliable across three groups (r(208) on shape similarity: 0.957–0.983; on conceptual association: 0.982–0.984), and the pairwise shape similarity was orthogonal to the pairwise conceptual association (r(208): 0.103–0.132).

We then averaged the pairwise rating scores of all the participants ($N = 48$) and calculated the representational dissimilarity matrix (RDM) of shape similarity and conceptual association (i.e., 7 minus the mean rating score). The resulting two model RDMs had comparable variance across pairs of objects (shape similarity: variance = 2.163; conceptual association: variance = 2.498) and therefore offered equated discovery possibilities when correlated with brain RDMs in the subsequent RSA. Fig 1C and 1D show the organizational structure of the two RDMs, where 21 items were grouped according to the clusters generated by the k-means clustering algorithm [38,39], with the silhouette criterion used to decide the optimal number of clusters [40]. The shape similarity RDM fell into three clusters, corresponding to square, round, and elongated objects (Fig 1C). The conceptual association RDM fell into seven smaller clusters, corresponding to different occasions in which objects were used (Fig 1D). For example, the two biggest clusters were related to eating and writing. The conceptual rating results accorded closely with the teleological perspective, which suggests the essence of a manmade object lies in its function, not its physical properties (e.g., [41]).

### Behavior rating on other object properties and confounding factors

Potential confounding factors were also considered. It has been reported that other properties of manmade objects can also modulate brain activity, like object size (big versus small; e.g., [42]), toolness (tools versus non-tool manmade objects; e.g., [43]), and contextual association

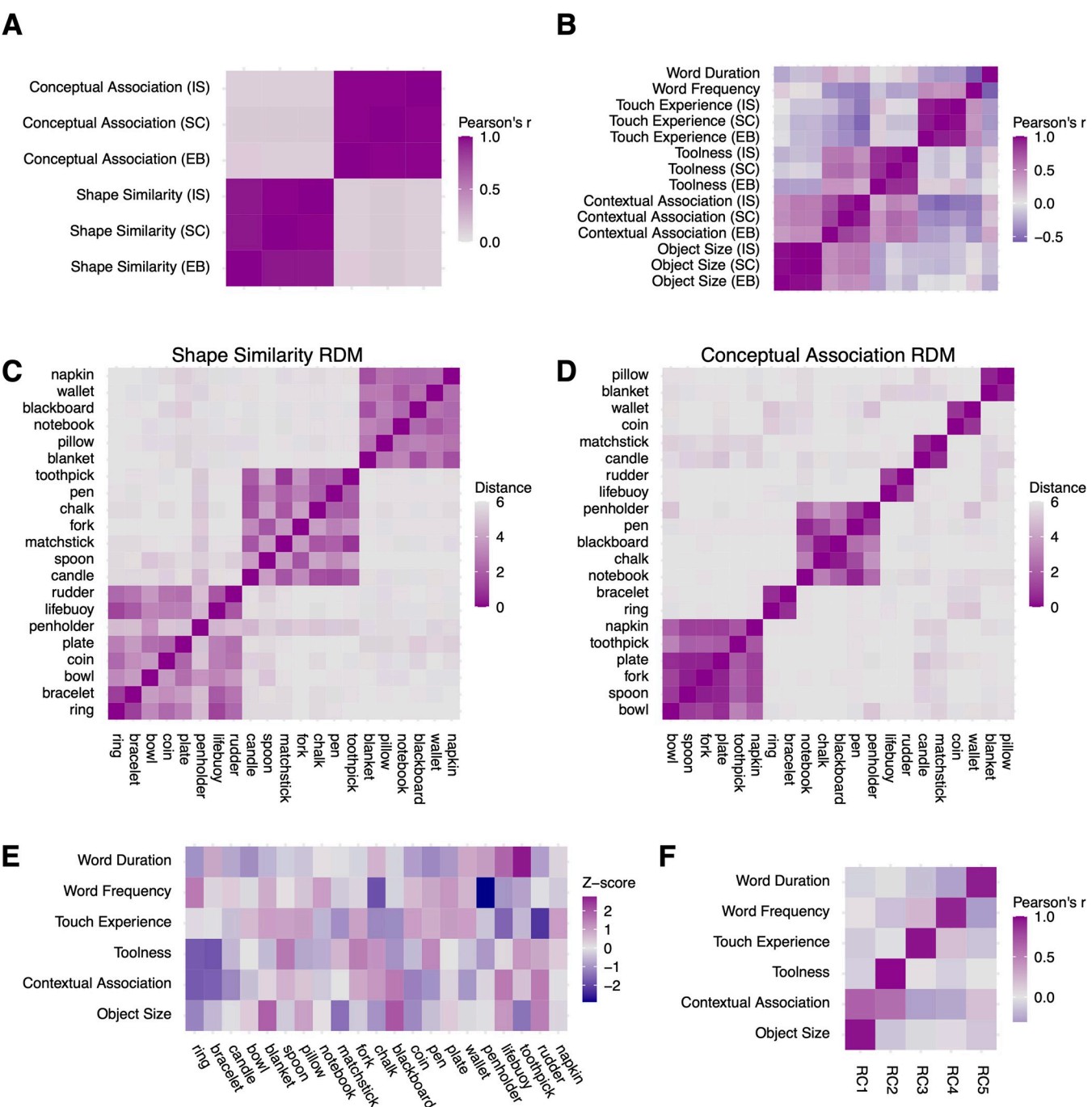

**Fig 1. Stimulus information.** (A) Correlation between ratings on pairwise shape similarity and pairwise conceptual association across three participant groups (EB: early blind, SC: sighted control, IS: independent sighted). (B) Correlation among linguistic variables and ratings on other object properties across three participant groups. (C) Pairwise ratings on shape similarity (i.e., the shape similarity RDM). (D) Pairwise ratings on conceptual association (the conceptual association RDM). (E) Linguistic variables and ratings on other object properties. (F) Correlations between the first five RCs and linguistic variables and ratings on other object properties. The underlying data for this figure can be found in S1 Data. RC, rotated component; RDM, representational dissimilarity matrix.

(strong versus weak contextual association objects; e.g., [44]). These three variables were rated on a 7-point Likert scale (see Procedures in Material and methods about the rating procedure). S3 Table shows the inter-rater reliability within each group of participants. The inter-rater

reliability reached "excellent" on object size (ICC(C,k): 0.979–0.992) and varied from "good" to "excellent" on toolness (ICC(C,k): 0.893–0.928). The inter-rater reliability on contextual association differed between sighted and early blind groups. While sighted groups had a "good" to "excellent" inter-rater reliability (SC: ICC(C,k) = 0.856; SI: ICC(C,k) = 0.919), the early blind group only had a "moderate" one (EB: ICC(C,k) = 0.613). Such heterogeneity in the early blind might result from a lack of instantaneous and global information about the environment from the visual input.

Besides the three object properties, all participants rated on a 7-point Likert scale about the degree to which they knew each object's typical shape and primary function. Since most stimuli selected were everyday objects, both shape and conceptual rating scores hit the ceiling and varied only slightly across objects (averaged shape familiarity score across objects: M = 6.744, SD = 0.285; averaged conceptual familiarity score across objects: M = 6.944, SD = 0.066). Participants also rated how frequently they touched each object (1: have never touched it before; 7: touch it every day), which can be considered a sensitive and common index reflecting object familiarity across sighted and early blind groups. The inter-rater reliability on touch experience within each group of participants reached "excellent" (ICC(C,k): 0.965–0.975; S3 Table).

We averaged the above rating scores within each group of participants and evaluated the reliability of the mean rating score across participant groups. Fig 1B shows that the rating scores across three groups of participants were reliable (r(19) on objects size: 0.973–0.998; on contextual association: 0.732–0.940; on toolness: 0.883–0.933; on touch experience: 0.935–0.974). From this figure, we can also spot a moderate positive correlation between object size and contextual association (r(19): 0.363–0.529) and between toolness and contextual association (r(19): 0.264–0.622), which means the bigger the size, or the more likely an object is a tool, the more likely this object is bound to a specific context. Moreover, we also added two linguistic measures—word frequency (i.e., the Zipf value of the word occurrence in film and television subtitles; http://crr.ugent.be/subtlex-it/) and word duration. There was a moderate positive correlation between word frequency and touch experience (r(19): 0.419–0.446) and a moderate negative correlation between word frequency and word duration (r(19) = −0.577).

We then averaged the rating scores across all participants (N = 48) to get a mean rating score vector for each rating item. Fig 1E illustrates the Z-scores of all the ratings across objects. To orthogonalize these unidimensional variables, we conducted the principal component analysis and applied varimax rotation to improve the interpretability of the resulting principal components. Five components had eigenvalues greater than 1. Fig 1F shows the correlation of these five rotated components (RCs) with each rating item. RC1 to RC5 corresponded to object size, toolness, touch experience, word frequency, and word duration, respectively (r(19): 0.915–0.981). The RCs corresponding to object size and toolness also had moderate correlations with the contextual association (r(19): 0.656 and 0.584). These RC scores were used in the subsequent parametric modulation analysis.

## Performance on shape and conceptual tasks during scanning

During the scanning, participants performed two tasks on the same set of auditorily presented words. In the shape verification task, participants thought carefully about the typical shape of each object and judged whether it was elongated, angular, hollow, circular, and disc-shaped. In the conceptual verification task, participants thought carefully about the primary function of each object and judged whether it was for eating, writing, sleeping, lighting, and purchasing (see S1 Fig and Procedures in Materials and methods for details).

Table 1 shows the accuracy and reaction time (RT) across participants within each group in shape and conceptual verification tasks. All groups of participants had near-ceiling accuracy

**Table 1. Accuracy and RT during fMRI scanning.**

| | Accuracy (%, M ± SD) | | RT (ms, M ± SD) * | |
|---|---|---|---|---|
| | Shape task | Conceptual task | Shape task | Conceptual task |
| EB (N = 16) | 95.0 ± 4.1 | 96.9 ± 1.6 | 1,798 ± 313 | 1,590 ± 234 |
| SC (N = 16) | 97.4 ± 2.7 | 98.1 ± 1.3 | 1,689 ± 265 | 1,546 ± 221 |
| IS (N = 16) | 95.9 ± 2.9 | 97.7 ± 1.5 | 1,999 ± 345 | 1,866 ± 376 |

* Mean RT across all the correct trials within each participant.

EB, early blind; fMRI, functional magnetic resonance imaging; IS, independent sighted; RT, reaction time; SC, sighted control.

on both tasks. The shape verification task took about 130 to 200 ms longer than the conceptual verification task. We built a linear mixed model to predict the RT in the correct trials with groups of participants (EB versus SC) and types of tasks (shape versus conceptual tasks) as fixed effects variables and each participant as random effects grouping factors. The analysis revealed a significant task effect ($F_{(1,30)} = 73.055$; $p < 0.001$), whereas the group effect was found insignificant ($F_{(1,30)} = 0.732$, $p = 0.399$), and the interaction effect between groups and tasks had only a slight trend toward significance ($F_{(1,30)} = 2.552$, $p = 0.123$). The significant difference between shape and conceptual tasks aligns with the evidence suggesting that retrieving specific semantic features (e.g., shape knowledge) requires more time than general semantic knowledge (i.e., function knowledge [45]). The interaction effect showed a weak trend that the shape task was slightly more difficult than the conceptual task for the EB than the SC ([(EB > SC) × (shape > conceptual tasks)]; $z = 1.588$, $p = 0.112$), which might be due to a lack of visual experience.

## Shape compared to conceptual tasks engaged ILOTC in both EB and SC

We first contrasted the neural activity level between the shape and conceptual tasks. To remove the domain-general RT effect, we modeled the trial-by-trial RT variability across the two tasks in the first-level general linear model (GLM) using both the variable epoch approach and the variable impulse approach [46]. Fig 2 illustrates results while the domain-general RT effect was controlled (vertex-wise $p < 0.001$, cluster-level family-wise error (FWE) corrected $p < 0.05$).

Fig 2A shows the contrast between the shape task and the conceptual task using all participants (N = 48). The shape task and the conceptual task involved dissociable brain networks. The shape task activated bilateral brain areas, including the ILOTC (i.e., the lateral part of the Broadman area (BA) 37), the aIPS, the posterior IPS (pIPS), the vPMC, and the inferior frontal sulcus. To verify whether the ILOTC activated in the shape task was the same region as the LOtv reported in previous literature, we projected the peak coordinates of the LOtv from three representative studies (i.e., [12,13,47]) to the brain surface and found that these coordinates largely fell over the geometric gravity center of the ILOTC region. In Fig 2A, we can identify two activity epicenters in the IPS—one was anterior and the other was posterior and joined to the intraoccipital sulcus.

The conceptual task mainly activated left-lateralized brain areas, including the anterior part of the lateral temporal lobe (aLTC), the superior temporal gyrus (STG; BA 22), the angular gyrus (AG; BA 39), and the supramarginal gyrus (SMG; BA 40). These regions were in accord with the high-level linguistic network [48–50] and are considered to underly language-supported conceptual processing [51–53].

We then looked at the brain activation in EB and SC separately (Fig 2B and 2C). Both EB and SC had ILOTC activation in the shape task compared to the conceptual task. To confirm that the regions in the ILOTC found in the two groups were the same, we calculated the

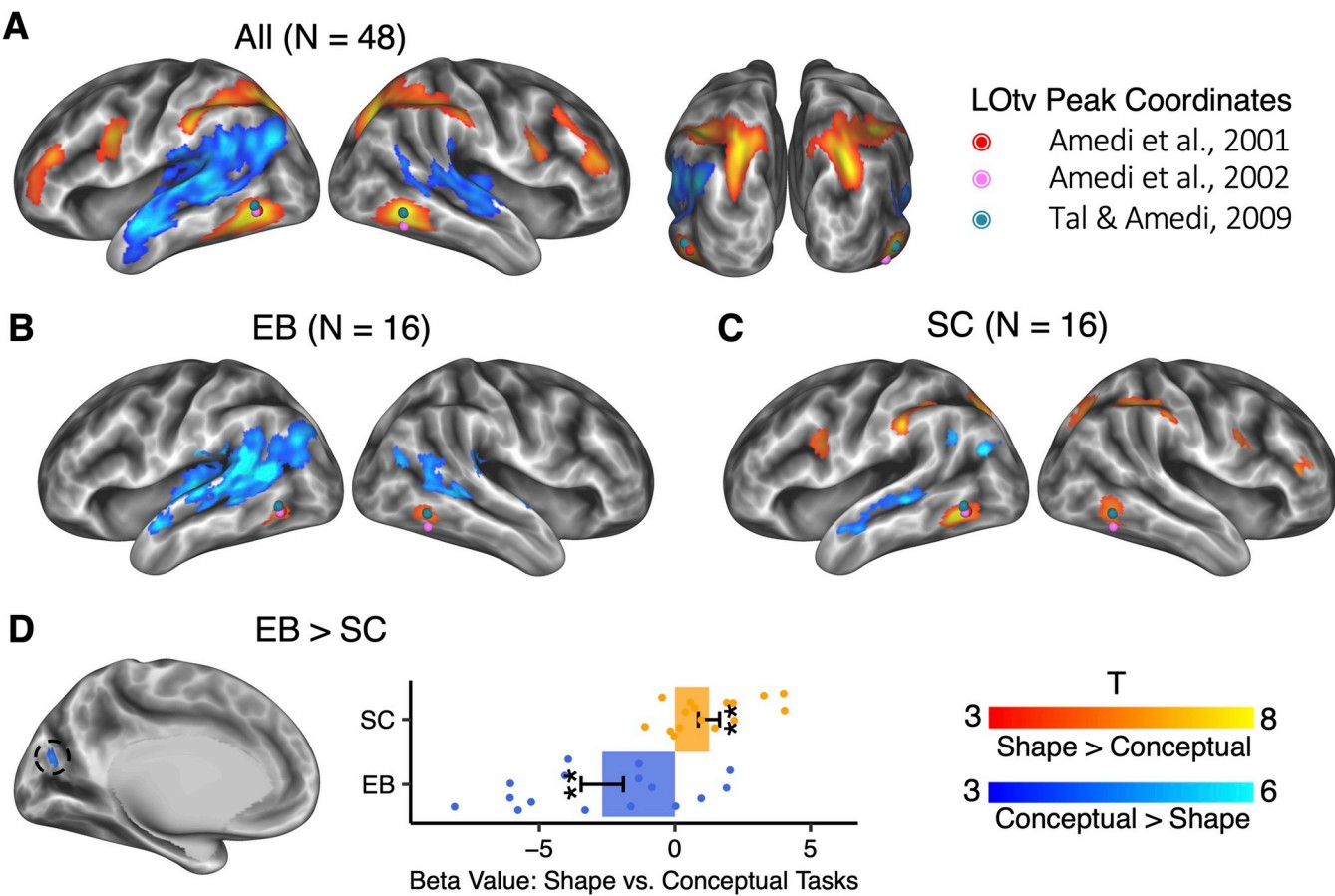

**Fig 2. Specific brain activation in shape and conceptual tasks (vertex-wise *p* < 0.001, cluster-level FWE corrected *p* < 0.05).** Dots in colors denote the location of LOtv in three representative studies. (A) Shape versus conceptual tasks across all the participants. (B) Shape versus conceptual tasks in the EB. (C) Shape versus conceptual tasks in the SC. (D) Interaction between groups (EB vs. SC) and tasks (shape vs. conceptual). The error bars indicate the standard error. **: *p* < 0.01. The underlying data for this figure can be found in S1 Data. EB, early blind; FWE, family-wise error; LOtv, lateral occipital tactile-visual; SC, sighted control.

overlap coefficient, i.e., the area of the intersection region divided by the smaller area of the two regions. The overlap coefficient of the left ILOTC was 100%, i.e., EB's ILOTC fell within SC's ILOTC. The overlap coefficient of the right ILOTC was 81.5%. Consistent with the results pooling all participants (*N* = 48), SC also had significant activation in bilateral aIPS, pIPS, and vPMC in the contrast between shape and conceptual tasks (Fig 2C). Although these regions did not survive the multiple comparison correction at the whole-brain level in EB (Fig 2B), analyses using the significant areas in SC as regions of interest (ROIs) showed bilateral aIPS, bilateral pIPS, and the left vPMC in the EB also showed greater activation in the shape task than in the conceptual task (S2 Fig; left aIPS: t(15) = 3.486, *p* = 0.003; right aIPS: t(15) = 2.487, *p* = 0.025; left pIPS: t(15) = 2.478, *p* = 0.026; right pIPS: t(15) = 3.357, *p* = 0.004; left vPMC: t (15) = 2.632, *p* = 0.019; right vPMC: t(15) = 1.861, *p* = 0.083).

Both EB and SC activated the language network in the conceptual task. However, EB exhibited reduced left lateralization than SC. To measure the extent of lateralization, we extracted the T scores of the top 5% percentage of vertices showing the strongest activation in the contrast between the conceptual task and the shape task within the language network, which was anatomically defined in each participant's native space by combing bilateral STG, bilateral

inferior parietal cortices (i.e., the AG), and bilateral SMG in the DKT atlas [54]. The left lateralization was measured as $(L - R)/(L + R)$, where L and R were the sums of T scores in the left and right hemispheres. While the SC had clear left lateralization (M = 0.381, SD = 0.344, t(15) = 4.440, $p < 0.001$), the EB's lateralization was not evident (M = 0.143, SD = 3.397, t(15) = 1.442, $p = 0.397$). The paired $t$ test showed a significant difference between the SC and the EB (paired t(15) = 2.452, $p = 0.027$), while no significant difference was found in handedness scores (SC: M = 76.875, SD = 20.238; EB: M = 73.750, SD = 16.279; paired t(15) = 0.543, $p = 0.595$). The reduced left lateralization for language processing in EB has been reported in a recent study and is still open to interpretation [55].

Next, we directly contrasted the neural activity between EB and SC. As a sanity check, we first compared the brain activity level in shape and conceptual tasks to the resting state between EB and SC. As both tasks included auditory input, the occipital cortex in EB should show enhanced activation due to cross-modal neuroplasticity (e.g., [56–59]), and the results showed up as expected (S3 Fig; vertex-wise $p < 0.001$, cluster-level FWE corrected $p < 0.05$). We then compared the activity level between shape and conceptual tasks between EB and SC. We found only one significant region in the left cuneus near the parieto-occipital sulcus (Fig 2D; vertex-wise $p < 0.001$, cluster-level FWE corrected $p < 0.05$). ROI analysis showed that this region in EB had greater activation in the conceptual task than in the shape task (t(15) = −3.447, $p = 0.004$), whereas, in SC, it showed an opposite pattern (t(15) = 3.213; $p = 0.006$). This finding suggests that the earlier "visual" cortex in EB (i.e., the left cuneus) might repurpose itself to a similar role as what the language network played in the conceptual task (see also a recent meta-analysis [60]).

S4A Fig illustrates the RT effect across the two tasks (N = 48; vertex-wise $p < 0.001$, cluster-level FWE corrected $p < 0.05$). As expected, it involved both frontoparietal and cingulo-opercular networks underlying top-down control [61]. It also involved regions in the default mode network, which could be because both the shape and the conceptual tasks require mental simulation [62]. Intriguingly, contrasting the RT effect between EB and SC revealed the lateral and ventral parts of the occipital cortex (S4B Fig; vertex-wise $p < 0.001$, cluster-level FWE corrected $p < 0.05$). These regions substantially overlapped with the LOC involved in visual shape perception in the sighted population, suggesting a functional reorganization of these regions in EB. Note that these regions did not overlap with the ILOTC.

## Other object properties did not modulate ILOTC activity

To investigate whether the other object properties modulated brain activity in the ILOTC, we conducted a parametric modulation analysis. The set of the parametric modulators included the task type (i.e., the shape task coded as 1 and the conceptual task coded as −1), the z-scores of the RT across all the trials in each run, the RCs corresponding to object size, toolness, touch experience, word duration, and word frequency. Fig 3 presents the significant brain areas encoding these parametric modulators (N = 48; vertex-wise $p < 0.001$, cluster-level FWE corrected $p < 0.05$).

When potential confounding factors were modeled, the difference between the two task types was still preserved (Fig 3A): The shape task activated bilateral brain areas, including the ILOTC, the aIPS, the pIPS, and the vPMC. The conceptual tasks mainly activated brain areas in the left hemisphere, including the orbital part of the inferior frontal gyrus (IFG) (i.e., BA 47), the aLTC, the posterior part of the STG (pSTG), the SMG, and the AG. These regions neatly matched the language network [48,49] with the absence of the triangular part of the IFG and the 55b region in the premotor cortex [63] (see the overlap in S5 Fig), in line with previous studies suggesting these two dorsal regions play a non-semantic role in language processing

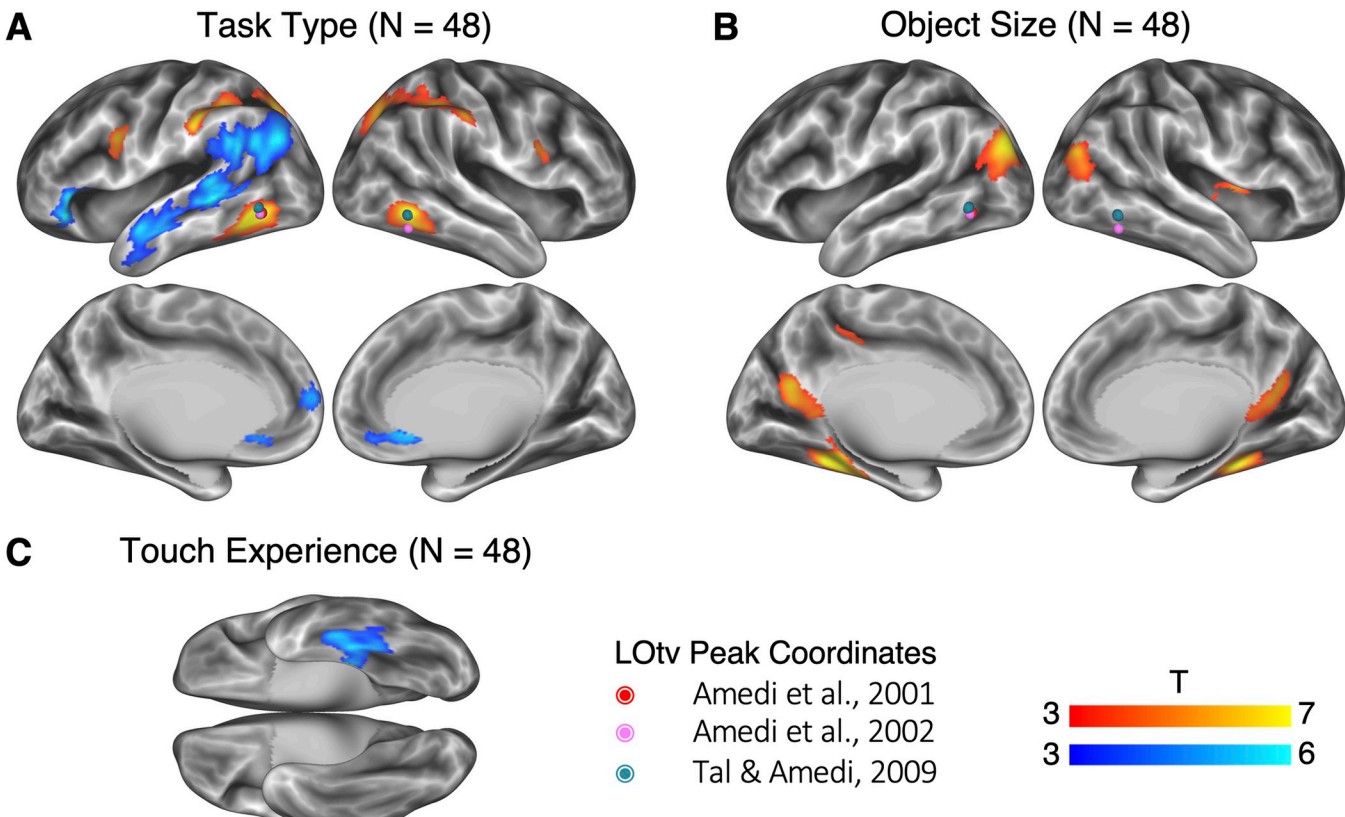

**Fig 3. Neural correlates of task types and other object properties (vertex-wise $p < 0.001$, cluster-level FWE corrected $p < 0.05$).** Dots in colors denote the location of LOtv in three representative studies. (A) Neural correlates of task types (the shape task coded as 1 and the conceptual task coded as −1). (B) Neural correlates of object size. Activations in the significant brain areas positively correlated with object size, i.e., larger objects induced higher activation. (C) Neural correlates of touch experience. Activations in the significant brain area negatively correlated with touch experience, i.e., less-touched objects induced higher activation. The underlying data for this figure can be found in S1 Data. FWE, family-wise error; LOtv, lateral occipital tactile-visual.

(e.g., [64–66]). Since the brain clusters in Fig 3A were more discrete than those reported in the univariate contrast reported in Fig 2 (with no control for alternative object properties), we used the significant regions in Fig 3A to define the ROIs in the following analyses. No regions showed significant differences between EB and SC. We also found the same region in the cuneus when directly comparing EB and SC under a lower threshold (vertex-wise $p < 0.001$, uncorrected).

Fig 3B and 3C shows the brain areas sensitive to the other object properties. The object size was mainly localized to the three scene-selective regions—the transverse occipital sulcus, the parahippocampal place area, and the retrosplenial cortex (Fig 3B). It has already been reported that these areas also prefer large nonmanipulable objects (e.g., [42,67]) and objects with a strong contextual association (e.g., [44,68]). Since the object size component here had a moderate correlation with the rating scores on contextual association (Fig 1F), we cannot distinguish between these two factors in this study. Moreover, we found a region in the left ventral and medial temporal cortex (mainly in the BA 20), of which the level of activity negatively correlated to touch experience (Fig 3C), suggesting this region was sensitive to the novelty of objects. We did not find any brain areas significantly modulated by toolness, which might result from the lack of typical tools (e.g., hammers or scissors) in the stimuli. Directly comparing the effects of all these parametric modulators between EB and SC also failed to reveal any significant brain regions.

S6 Fig illustrates the effect of the two linguistic variables ($N = 48$; vertex-wise $p < 0.001$, cluster-level FWE corrected $p < 0.05$). Word duration was localized to bilateral auditory cortices and bilateral STG. Word frequency was mainly localized to the right-lateralized ventral attention network and the salience network, characterized by their sensitivity to salient stimuli (e.g., [69,70]).

## ILOTC represented shape similarity, not conceptual association in both EB and SC

We then used RSA to investigate whether the ILOTC identified implemented shape representation (Fig 4, left and right panels corresponding to left and right ILOTC). A three-way mixed ANOVA was first performed between groups (EB versus SC), tasks (shape versus conceptual tasks), and representations (shape similarity versus conceptual association). The groups factor was between-subject, whereas tasks and representations were within-subject factors. In bilateral ILOTC, we only found a significant effect in representations and a significant interaction between tasks and representations (Table 2).

Fig 4A illustrates the RSA results in bilateral ILOTC across all participants ($N = 48$). Bilateral ILOTC represented shape similarity in both the shape task (left ILOTC: t(47) = 10.367, $p < 0.001$; right ILOTC: t(47) = 7.705, $p < 0.001$) and the conceptual task (left ILOTC: t(47 = 4.066), $p < 0.001$; right ILOTC: t(47) = 3.209, $p = 0.002$). The shape representation was stronger in the shape task than in the conceptual task (left ILOTC: paired t(47) = 5.183, $p < 0.001$; right ILOTC: paired t(47) = 3.776, $p < 0.001$). We found no clear evidence that bilateral ILOTC represented the conceptual association in either the shape or the conceptual tasks—only the conceptual effect in the left ILOTC in the conceptual task was marginally significant (t(47) = 2.123, $p = 0.039$). No significant difference was found in conceptual representation between shape and conceptual tasks (left ILOTC: paired t(47) = 0.558, $p = 0.580$; right ILOTC: paired t(47) = 0.395, $p = 0.695$).

Fig 4B highlighted that the population without visual experience (i.e., the EB) showed a largely similar pattern. Bilateral ILOTC represented shape similarity in the shape task (left ILOTC: paired t(15) = 4.568, $p < 0.001$; right ILOTC: paired t(15) = 3.610, $p = 0.003$), whereas their shape representation in the conceptual task was less evident (left ILOTC: paired t(15) = 1.220, $p = 0.241$; right ILOTC: paired t(15) = 1.852, $p = 0.084$). The paired $t$ test revealed a significant difference between the two tasks in the left ILOTC (paired t(15) = 3.361, $p = 0.004$) but not in the right ILOTC (paired t(15) = 1.466, $p = 0.163$). No evidence supported bilateral ILOTC represented conceptual association in either shape or conceptual tasks (t(15) < 1.282, ps > 0.219).

We also investigated whether bilateral ILOTC in EB and SC share a matched shape representation (Fig 4C). By doing so, we measured the within-group coherence—the correlation between each participant's neural RDM and the mean neural RDM of the other participants within the same group (i.e., EB-EB and SC-SC) and the between-group coherence—the correlation between each participant's neural RDM and the mean neural RDM of all the other participants in the other group (i.e., EB-SC). A two-way ANOVA was performed between tasks (shape versus conceptual tasks) and group pairs (EB-EB versus SC-SC versus EB-SC). No significant interaction was found between tasks and group pairs (left ILOTC: F(2,90) = 1.366, $p = 0.260$; right ILOTC: F(2, 90) = 1.446, $p = 0.241$). There is a significant difference between tasks (left ILOTC: F(1,90) = 90.743, $p < 0.001$; right ILOTC: F(1, 90) = 75.809, $p < 0.001$), suggesting the shape task induced more coherent representations in bilateral ILOTC across participants. A weak effect in group pairs was also spotted in the left ILOTC (F(2,90) = 4.746, $p = 0.011$) but not in the right one (F(2, 90) = 1.065, $p = 0.349$). The post hoc comparison

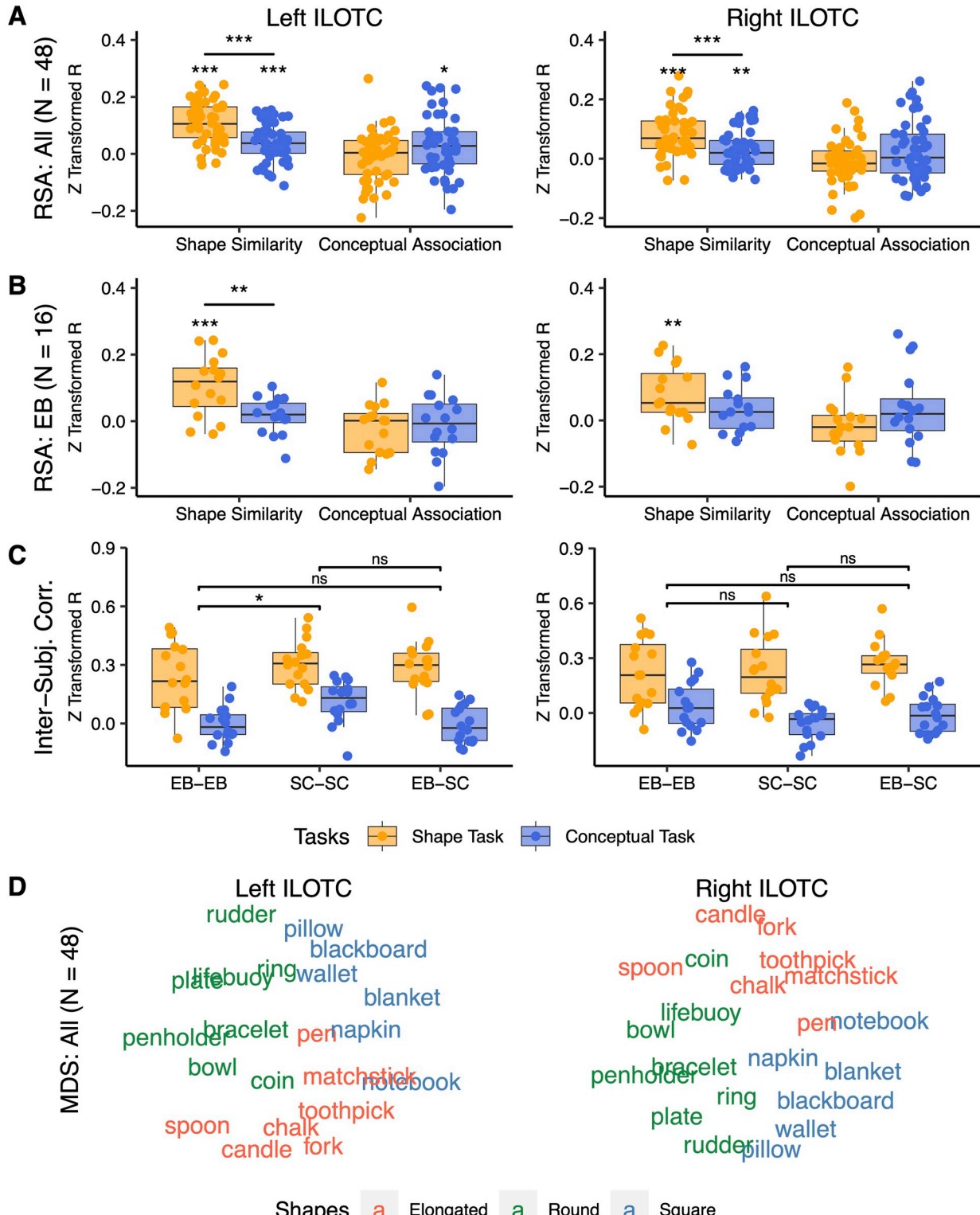

**Fig 4. Neural representations of bilateral ILOTC.** The left column showed the neural representation in the left ILOTC. The right column showed the neural representation of the right ILOTC. (A) The RSA results across all participants ($N = 48$). (B) The RSA results in the EB ($N = 16$). (C) Inter-subject correlation between brain RDMs within and between the EB and the SC. (D) The MDS visualization of the mean brain RDM of the ILOTC across all participants ($N = 48$). ns: not significant, *: $p < 0.05$, **: $p < 0.01$, ***: $p < 0.001$. The underlying data for this figure can be found in S1 Data. EB, early blind; ILOTC, inferolateral occipitotemporal cortex; RDM, representational dissimilarity matrix; RSA, representational similarity analysis; SC, sighted control.

**Table 2. Neural representation in bilateral ILOTC.**

| Three-way mixed ANOVA * | Left ILOTC | | Right ILOTC | |
|---|---|---|---|---|
| **Groups** (EB vs. SC) | F(1, 30) = 1.809 | $p = 0.189$ | F(1, 30) = 0.466 | $p = 0.500$ |
| **Tasks** (shape vs. conceptual) | F(1, 30) = 2.829 | $p = 0.103$ | F(1, 30) = 0.025 | $p = 0.874$ |
| **Representations** (shape vs. conceptual) | **F(1, 30) = 21.814** | **$p < 0.001$** | **F(1, 30) = 11.871** | **$p = 0.002$** |
| **Groups × Tasks** | F(1, 30) = 2.047 | $p = 0.163$ | F(1, 30) = 0.157 | $p = 0.695$ |
| **Groups × Representations** | F(1, 30) = 2.056 | $p = 0.162$ | F(1, 30) = 0.055 | $p = 0.816$ |
| **Tasks × Representations** | **F(1, 30) = 15.596** | **$p < 0.001$** | **F(1, 30) = 10.116** | **$p = 0.003$** |
| **Groups × Tasks × Representations** | F(1, 30) = 0.097 | $p = 0.757$ | F(1, 30) = 0.066 | $p = 0.799$ |

* The groups factor was between-subject, whereas tasks and representations were within-subject factors.

EB, early blind; ILOTC, inferolateral occipitotemporal cortex; SC, sighted control.

found that the mean value across levels of tasks was significantly different between SC-SC and EB-EB in the left ILOTC (Tukey's test: $p = 0.011$), suggesting that the neural representation in the left ILOTC was more homogeneous in the SC group than in the EB group. However, there was no significant difference between EB-EB and EB-SC (Tukey's test: $p = 0.742$) or between SC-SC and EB-SC (Tukey's test: $p = 0.073$), suggesting no significant evidence showing a boundary effect between the neural representations across groups.

We averaged the neural RDMs of bilateral ILOTC across all participants ($N = 48$) and provided a planar visualization of the representational pattern using multidimensional scaling (Fig 4D). The color of words denoted the three clusters in the model RDM of shape similarity, mainly corresponding to elongated, round, and square objects. Representations of the three shape categories were separated in bilateral ILOTC.

We also investigated the multivariate object representation in other regions showing an enhanced univariate response to the shape task than the conceptual task. S4 to S6 Tables show the three-way mixed ANOVA results between groups (EB versus SC), tasks (shape versus conceptual task), and representations (shape similarity versus conceptual association) in bilateral aIPS, bilateral pIPS, and bilateral vPMC, respectively. They all had the same pattern, with a significant effect in representations and a significant interaction between tasks and representations. S7 Fig shows that all these regions represented shape similarity in the shape tasks (t(47): 5.531–10.074, ps < 0.001). Bilateral aIPS and pIPS also represented shape similarity in the conceptual tasks (t(47): 2.216–2.902, ps: 0.032–0.006), whereas shape representation in bilateral vPMC was not evident in the conceptual task (left: t(47) = 1.875, $p = 0.067$; right: (47) = 1.677, $p = 0.100$). Shape representation was more apparent in the shape task than in the conceptual task in all these regions (right vPMC: paired t(47) = 2.602, $p = 0.012$; other regions: paired t (47): 0.408–5.055, ps < = 0.001).

S8A Fig illustrated the whole-brain searchlight results of shape similarity in the shape tasks across all participants ($N = 48$) (vertex-wise FWE corrected $p < 0.005$, cluster size > 400 mm$^2$). The ILOTC was one of the epicenters showing the strongest shape effect. Direct contrast between EB and SC revealed a region in the right lateral occipital cortex showing a stronger shape representation in the EB than SC (S8B Fig; vertex-wise $p < 0.001$, cluster-level FWE corrected $p < 0.05$).

### Conceptual representation in the brain

We also used the RSA to investigate whether the brain areas sensitive to the conceptual task in the univariate analyses represented multivariate conceptual association (Fig 5). Interestingly, although all these regions showed significantly stronger univariate activation in the conceptual

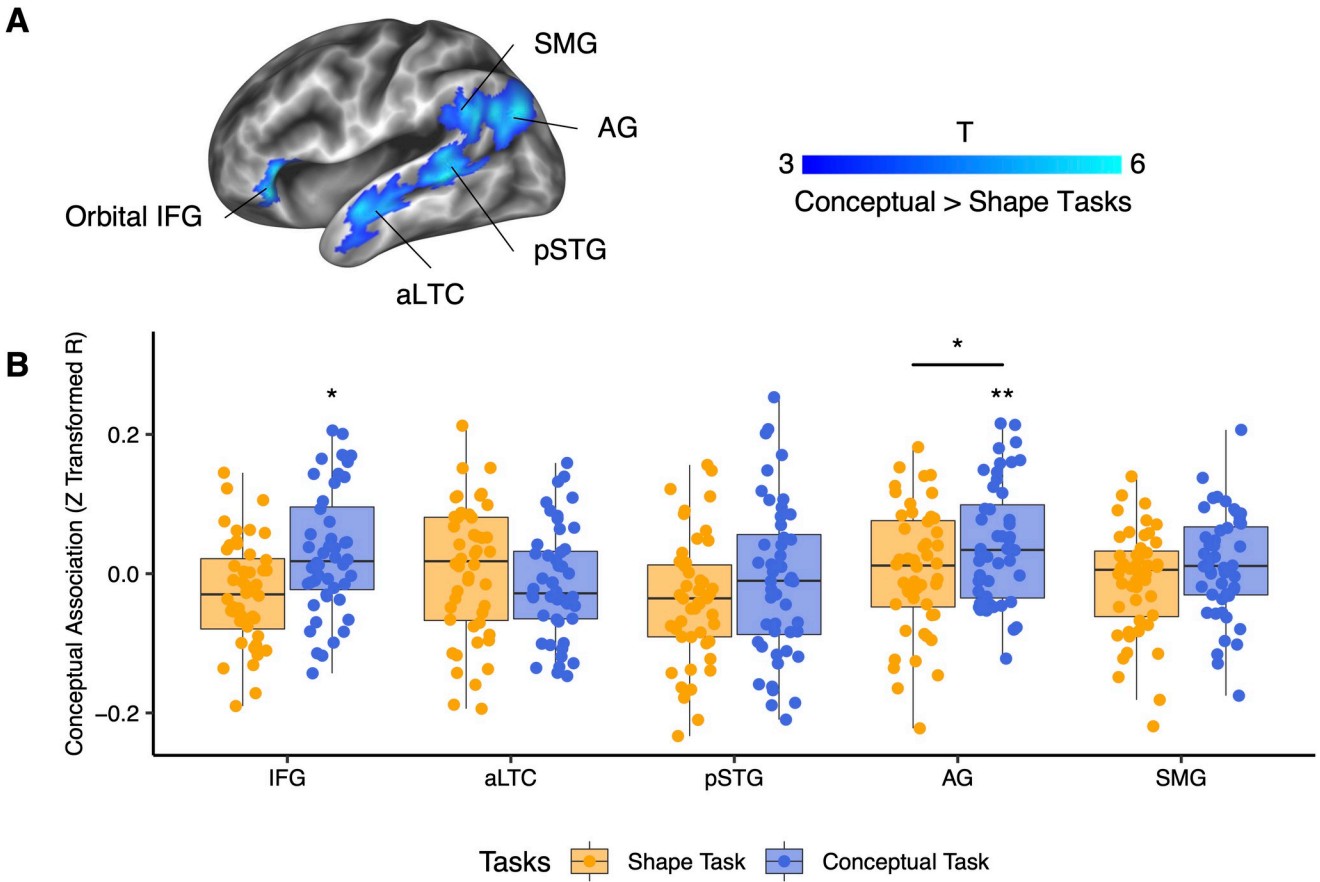

**Fig 5. RSA results of conceptual association in the brain areas with greater activation in the conceptual task than in the shape task across all participants (N = 48).** (A) Brain areas with significantly greater activation in the conceptual task than in the shape task defined in Fig 2A. (B) RSA results of these conceptual-relevant areas in shape and conceptual tasks. *: $p < 0.05$, **: $p < 0.01$. The underlying data for this figure can be found in S1 Data. RSA, representational similarity analysis.

task than in the shape task, only the left AG represented the conceptual association in the conceptual task across all the participants (orbital IFG: t(47) = 2.395, $p = 0.021$; aLTC: t(47) = −1.268, $p = 0.211$; pSTG: t(47) = −0.621, $p = 0.537$; AG: t(47) = 3.337, $p = 0.002$, SMG: t(47) = 1.174, $p = 0.246$; only the AG survived from multiple comparison correction, as Bonferroni corrected $p < 0.05$ for five ROIs is $p < 0.01$). The conceptual representation in the left AG was more evident in the conceptual task than in the shape task (paired t(47) = 2.163, $p = 0.036$), and no group differences were found between EB and SC (F(1, 30) = 0.192, $p = 0.664$).

S9 Fig illustrates the whole-brain searchlight results of conceptual association in the conceptual tasks across all participants (N = 48) (vertex-wise $p < 0.001$, cluster-level FWE corrected $p < 0.05$). The effects were mainly on bilateral dorsal AG, the left pIPS, the left precuneus, and the left dorsal medial prefrontal cortex. Given that some of the regions could also be spotted in the shape effect in the shape task (S8A Fig), they were likely to be driven by the task context [71].

## Shape and conceptual brain network in both EB and SC

We last used the seed-based RSFC to trace the regions having the neural activity synchronized with bilateral ILOTC (left ILOTC: Fig 6A; right ILOTC: Fig 6B; vertex-wise FWE corrected

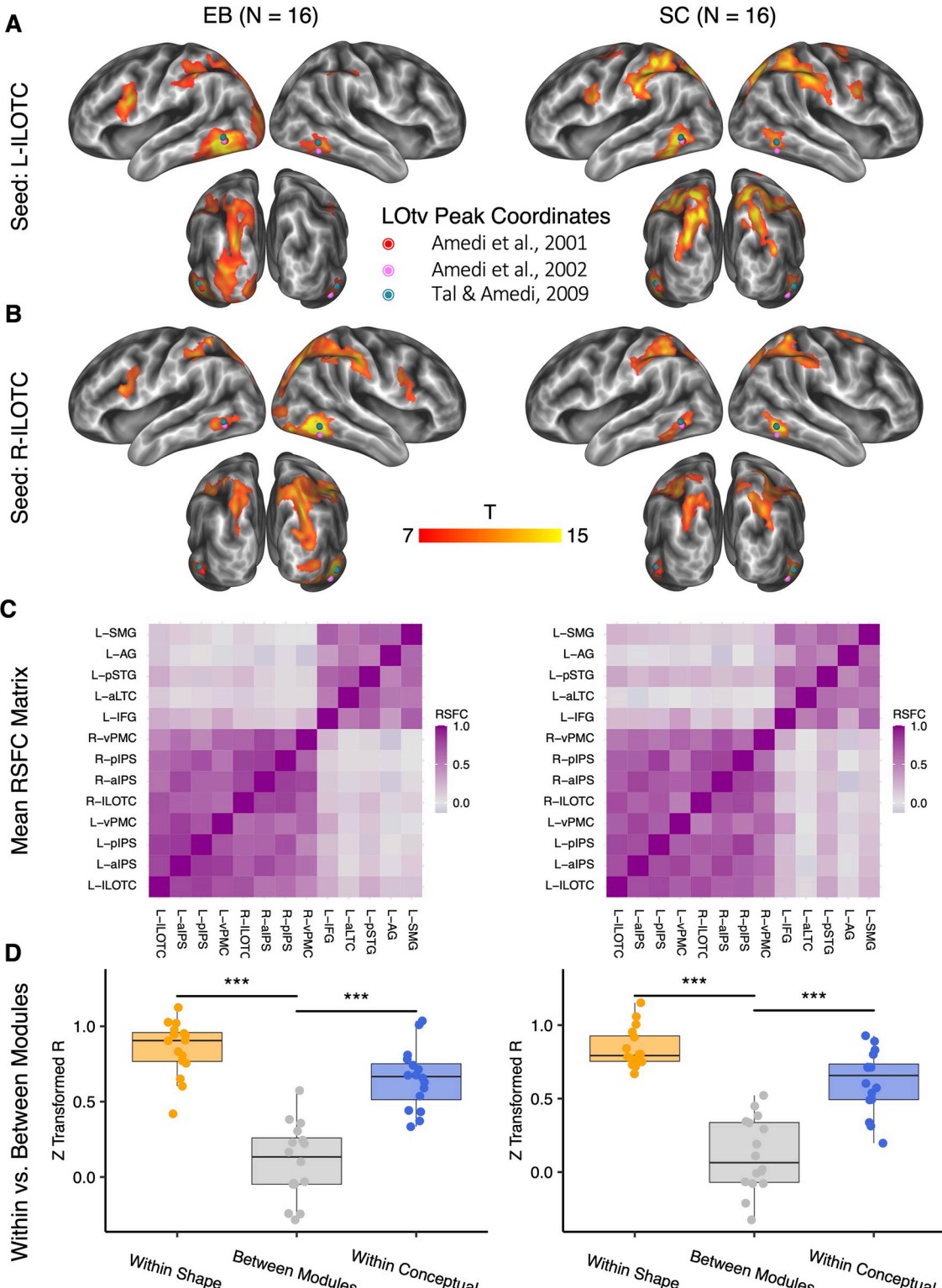

**Fig 6. Shape and conceptual brain network.** The left panel shows the RSFC results in the EB, and the right panel shows the RSFC results in the SC. (A, B) The significant seed-based RSFC results in the left ILOTC (A) and the right ILOTC (B) (vertex-wise FWE corrected $p < 0.005$, cluster size > 400 mm²). Dots in colors denote the location of LOtv in three representative studies. (C) The mean RSFC matrix across participants in EB and SC among the shape- and conceptual-relevant brain areas. (D) Comparison among the mean RSFC among the shape-relevant regions ("Within Shape"), among the conceptual-relevant regions ("Within Conceptual"), and

between the shape- and the conceptual-relevant regions ("Between Modules"). ***: $p < 0.001$. The underlying data for this figure can be found in S1 Data. EB, early blind; FWE, family-wise error; ILOTC, inferolateral occipitotemporal cortex; LOtv, lateral occipital tactile-visual; RSFC, resting-state functional connectivity; SC, sighted control.

$p < 0.005$, cluster size $> 400$ mm$^2$). The ILOTC had strong RSFC to the other bilateral regions sensitive to the shape task—the aIPS, the pIPS, and the vPMC in both EB and SC. The left ILOTC in EB had stronger connectivity to the "visual" cortex than in SC (S10 Fig; vertex-wise $p < 0.001$, cluster-level FWE corrected $p < 0.05$).

Fig 6C illustrates the mean RSFC matrix across participants in EB and SC among the regions showing stronger activation in the shape tasks or in the conceptual task. It shows that the brain areas sensitive to the shape task and those sensitive to the conceptual task belonged to separate network modules in both EB and SC. Fig 6D further compares the mean RSFC across all the pairs among the shape-sensitive regions, among the conceptual-sensitive regions, and between the shape- and the conceptual-sensitive regions. In both EB and SC, the mean RSFC within the shape module (EB: paired t(15) = 10.650, $p < 0.001$; SC: paired t(15) = 9.563, $p < 0.001$) and within the conceptual module (EB: paired t(15) = 10.024, $p < 0.001$; SC: paired t(15) = 8.014, $p < 0.001$) were significantly stronger than the mean RSFC between the two network modules.

## Discussion

Our study investigated where and how shape representations are stored in the brain and distinguished from the conceptual representation of the same manmade objects. By testing early blind participants, we assessed whether occipital regions implement shape representation independently of visual experience/imagery (e.g., [1–3]) or, alternatively, whether the "visual" cortex would repurpose its function for conceptual representation due to early visual deprivation [32]. We found that bilateral ILOTC, a region that overlaps with the LOtv [12,13,47], together with bilateral aIPS, pIPS, and vPMC, showed greater activation when people processed shape rather than conceptual attributes of the same objects, and their activity pattern encoded shape similarity but not conceptual association among objects. In contrast, regions in the left perisylvian area, including the orbital IFG, the aLTC, the pSTG, the AG, and the SMG, showed greater activation in the conceptual task than in the shape task. RSFC analysis further demonstrated that shape- and conceptual-relevant regions formed distinct brain networks. Interestingly, in all the above results, visual experience had little influence—EB and SC had similar activity profiles and connectivity patterns.

Our results thus favor the hypothesis suggesting the ILOTC implements supramodal shape representation and argue against the alternative hypotheses that such activation depends on visual imagery or conceptual associations based on functional relevance. These results echoed various perspectives suggesting object representation in the brain is organized according to properties, not modalities (e.g., [3,72,73]).

In contrast to the view that ILOTC implements supramodal shape representation, one could argue that this region might represent visual shapes in the sighted and haptic shapes in the early blind. Testing this possibility using fMRI is challenging as it is difficult to distinguish supramodal representation and visual representation derived from visual imagery triggered by touch in the sighted participants. One option would be to examine whether sighted patients with bilateral lesions in the ILOTC have both visual and tactile shape agnosia or only visual shape agnosia. Unfortunately, the two existing cases of bilateral ILOTC lesions cannot convincingly answer this question. One case is patient D.F., who had bilateral lesions in the LOC [10] and had both visual and tactile agnosia [74]. However, D.F. also had bilateral lesions to

the parieto-occipital cortex [75] and her tactile agnosia might result from parietal damage. The other case is patient M.C., who had bilateral lesions in the LOC, including the LOtv [15]. Unlike D.F., M.C. only had visual agnosia, and her tactile recognition ability was fast and accurate. However, although the haptic shape task did not activate the ILOTC of M.C. due to lesions in this region, it activated a nearby region in the posterior middle temporal gyrus. Such activation might reflect post-lesion reorganization, compensating for the shape representation that should be implemented in the ILOTC [15]. Besides resorting to rare patient cases, another seemingly plausible option would be selective transcranial magnetic stimulation (TMS) over bilateral ILOTC to evaluate whether it interferes with both haptic and visual shape tasks or only visual shape tasks. However, the pitfall is that even if TMS over bilateral ILOTC does disrupt haptic shape tasks (e.g., longer RT), such disruption might be mediated by the disruption in visual imagery, a strategy sighted participants would adopt to facilitate haptic shape tasks (e.g., [21,76]).

While conclusive proof is still warranted, there is other evidence supporting the role of ILOTC in supramodal shape representation in the sighted population. On the one hand, the ILOTC (mainly in BA 37) is anterior to the lateral occipital cortex (LO, mainly in BA 18), a visual shape perception region representing shape features like curvatures and medial axes, in contrast to the earlier visual cortex implementing retinotopic representation (e.g., silhouettes) (e.g., [77,78]). According to embodied semantic theories [79] and the "anterior shift" phenomenon [80], the associative cortex anterior to each sensorimotor area can gradually capture the regularities of the activity patterns in its nearby sensorimotor cortices induced by different exemplars of the same concept (e.g., different exemplars of an apple) and generate a schema-like representation as the sensorimotor knowledge of that concept (e.g., the typical color, shape, and action related to an apple). In line with this hypothesis, previous studies have shown that the region representing objects' color knowledge is localized to the fusiform gyrus anterior to the color perception area in V4 [81,82], and language-induced category-specific activations are aligned with but anterior to the visual-induced activations of the same semantic category [83]. The ILOTC, which is anterior to the LO, thus possibly represents objects' shape knowledge—the schematic or prototypic shape of an object concept—derived from various concrete shape exemplars represented in the LO of the sighted people.

On the other hand, the ILOTC was strongly connected to the IPS and the vPMC (Fig 6), a frontoparietal circuit that has long been proposed to be involved in hand configuration to grasp objects in particular shapes in light of single-neuron recording evidence (see reviews [84,85]). Neuropsychological evidence confirms that lesions in the aIPS can induce both tactile shape agnosia [16] and tactile apraxia [17], and lesions in the vPMC can lead to syndromes resembling tactile apraxia [18]. Our study found that the IPS-vPMC circuit implemented shape representation even in the early blind population with no visual experience (S2 Fig, S4–S6 Tables), further demonstrating that haptic sources alone can form the shape representation in these regions.

Converging the two groups of evidence described above—the position in the ventral visual pathway and the connection to the frontoparietal haptic circuit, it appears parsimonious to postulate that the ILOTC act as an operator bridging visual and haptic shape representations. Given this supramodal nature, the ILOTC might not only schematize the visual shape representation from the LO but also integrate the haptic shape representation from the IPS-vPMC circuit by amplifying the "affordance" shape features utilized for object grasping. This hypothesis is supported by the evidence that the ILOTC is more sensitive to pictures of graspable tools over non-graspable manmade objects (e.g., [86]), and its activity pattern better reflects the shape of objects' handles than their functional parts [87]. Nevertheless, the shape representation in the ILOTC is essentially sensorimotor-derived and would still be in the analogical

format, in contrast to the amodal symbolic format usually supported by the language system (e.g., the symbol of a "ring" associated with the symbol of "round"; similarly as the way the early blind represent color knowledge [53,88–90]).

As for the neural representation of functional knowledge, contrasting the conceptual task with the shape task revealed the left perisylvian regions related to linguistic processing (Figs 2 and 3), implying that function knowledge is supported by the language system. This result is supported by a recent massive study with 136 acute left hemisphere stroke patients [91]. They found that the deficit in tool selection (e.g., choosing the nail for the hammer) was specifically related to lesions in the left perisylvian regions, mainly including the whole length of the lateral temporal lobe and the anterior IFG. The language system might provide a symbolic format of representations, which can better capture the abstract "associations" among holistic concepts. It contrasts with the analogical format of representation grounded in the sensorimotor system (as discussed for the shape representation in ILOTC), which can better reflect the "similarity" in one particular semantic feature. Such findings suggest that function is not an explicit object property, which can be directly derived from sensorimotor experience—we cannot reduce an object's function to what it looks like and how it is manipulated; it must therefore rely on some sort of abstract/linguistic coding.

The differences between these two neural coding mechanisms may explain the discrepancy in the RSA results between shape similarity and conceptual association. RSA assumes that the representational content can be inferred from the distributed activity pattern across cortical surfaces. The most definitive evidence supporting this assumption comes from the primary sensorimotor system following a topographic organization (e.g., retinotopy), where the input and output information is transparently projected to the cortical surface. Since the shape representation (e.g., curvatures and medial axes) is transited and abstracted from the topographic representation [77,78], the activity pattern across the cortical surface in the shape-relevant regions would still be informative. However, in the linguistic system, the representation is presumed to be coded in the format of "arbitrary" symbols, where the linguistic sign (e.g., word forms) bears no obvious resemblance to the content signified. The content represented in the language system thus is not directly transited or abstracted from the word form representations in the sensorimotor cortex and might not be transparently reflected on the activity pattern across the cortical surface. Our results confirmed this hypothesis. Whereas all the shape-relevant regions defined by the univariate contrast encoded the shape similarity among objects (Figs 4 and S7), among the conceptual-relevant regions defined by the univariate contrast (i.e., aLTC, pSTG, SMG, AG, and orbital IFG), only the activity pattern in the AG was correlated to the conceptual association RDM (Fig 5).

The exception for the AG indicates that conceptual associations can be represented in a format other than linguistic. Previous studies have shown that the AG is not a purely linguistic region but also part of the default mode network engaged in memory-based simulation (e.g., [51,52,62,92]). Compared to the other linguistic regions, the AG is less responsive to word forms (e.g., [93]) but more sensitive to the retrieval of multimodal episodic memories (e.g., see review [94]). It is thus possible that the AG codes thematic relations based on the spatiotemporal continuity in our sensorimotor experience (e.g., hammers and nails often co-occur; e.g., [95]), which is apt to reflect on activity patterns [96], in contrast to the coding based on linguistic associations in the other language areas. In line with this idea, we found other brain regions in the default mode network also representing conceptual associations in the whole-brain searchlight RSA analysis, including the left precuneus and the left dorsal medial prefrontal cortex (S9 Fig).

It is worth noting that this study only focused on one type of conceptual knowledge: thematic relations or conceptual associations based on functional knowledge ("plate" and "fork"

used for dining versus "pillow" used for sleeping). The conclusion, therefore, cannot extend to other knowledge types like taxonomic categories ("plate" and "pillow" as manmade objects versus "dog" as animals). In the "Stimuli" section, we elaborated on why we narrowed down the stimuli to one taxonomic category (i.e., the manmade objects) and only focused on thematic relations. One of the main reasons is that it is challenging for a neuroimaging study to distinguish whether a brain area represents taxonomic knowledge per se or simply shows a preference for the features of specific categories. Take the ILOTC as an example. Previous studies have shown that regions overlapping or superior to the ILOTC are more sensitive to manmade objects than the other categories, even in early blind participants (e.g., [25–28]). However, this does not necessarily mean the ILOTC represents taxonomic knowledge at the conceptual level. Instead, our results suggest that the ILOTC represents shape knowledge derived from both visual and haptic modalities, and its preference for manmade objects is likely perceptual. Either because people have more haptic experiences with manmade objects or the manmade objects have more affordance shape features for grasping, the ILOTC can receive additional shape information from haptic modalities and therefore become more sensitive to the shapes of manmade objects (see previous discussions).

Our study also reveals crucial neuroplastic principles about how the "visual" cortex reorganizes its function after vision loss. In the high-order visual cortex, where brain areas receive not only visual input but also information from other sensorimotor systems, brain functions are likely to be resilient to vision loss through compensation. The most well-documented example is the region hMT+/V5, a highly specialized area for visual motion processing. This region also has a direct white matter connection to the planum temporale specialized in auditory motion processing [97] and preferentially responds to moving auditory and tactile stimuli in the early blind (e.g., [98–102]). Our results reveal the ILOTC has a similar nature—it had strong connections to the frontoparietal regions involved in haptic processing and preserved its functionality despite the lack of visual input (Figs 2 and 4).

In contrast, in the more primary visual cortex, where visual input is dominant, vision loss will leave a functional vacancy that would be difficult for another sense to fill in. Higher-order brain systems might have the opportunity to take over, pushing for a more radical functional repurposing in those early visual regions. This hypothesis is supported by neuroimaging studies showing that part of the "visual" cortex of the early blind is sensitive to linguistic components (semantics and syntax; e.g., [34,60,103,104]) and mathematical difficulty [105]. In line with these findings, we found that the left cuneus in EB showed greater activation to the conceptual task than the shape task, whereas the same conceptual preference can only be observed in the left perisylvian language areas in SC (Fig 2). Similarly, the lateral occipital cortex and the posterior fusiform gyrus in EB—two "earlier" regions along the visual processing stream than the ILOTC—showed a domain-general RT effect, which is typically observed in the frontoparietal and cingulo-opercular areas in SC (S3 Fig).

To conclude, our study identified dissociable brain networks representing objects' shape and conceptual knowledge. The bilateral ILOTC-IPS-vPMC circuit represented shape knowledge, and the left perisylvian circuit related to language processing represented conceptual knowledge. Relying on data collected in EB, we highlighted that the ILOTC represented shape knowledge independently of visual experience. We argue that the ILOTC implements a supramodal shape representation by virtue of its position in the ventral visual pathway and its strong connections to the IPS-vPMC circuit involved in haptic processing, and such sensorimotor-derived representation differs from the disembodied representation supported by the language system in their representational formats.

## Materials and methods

### Ethics statement

This study was conducted according to the principles expressed in the Declaration of Helsinki. The ethical committee of the University of Trento approved the experimental protocol in this study (protocol 2014–007). All participants provided written informed consent and were paid for their time.

### Participants

Forty-eight native Italian speakers with no history of neurobiological or psychiatric disorders participated in the fMRI experiment. Thirty-two participants were sighted and 16 participants were early blind. Further recruitment of blind participants was stalled due to COVID-19 restrictions. The early blind (EB) group reported, at most, faint light perception and had no visual memories (10 females; age: M = 32.8, SD = 4.5; all right-handed). To match the demographic information of the early blind group, we divided the sighted participants into two groups. Sixteen formed the sighted control (SC) group, matching the early blind in pairs on gender and age (10 females; age: M = 32.5, SD = 5.9; all right-handed). There was no significant difference between the early blind and the sighted control in head motion measured by the mean framewise displacement index [106] (EB: M = 0.20 mm, SD = 0.05 mm; SC: M = 0.17 mm, SD = 0.06 mm; t(30) = 1.79, $p$ = 0.083). The other 16 formed the independent sighted (IS) group (7 females; age: M = 28.3, SD = 8.1; 2 left-handed). We investigated the group-general effect by pooling EB, SC, and IS together to increase the statistical power and provide the most stable results. We investigated the between-group difference by contrasting EB and its matched SC.

S1 Table shows the demographic information of the early blind and their matched sighted control. In each matched pair, the gender was the same, and the age difference was no more than 3 years. All blind participants were blind since birth except for three participants, who also had visual trouble since birth but fully lost their vision at 8 months, 2 years, and 4 years. These participants' data did not differ from those of the other blind participants.

### Stimuli

To disentangle shape and conceptual representation, we aimed to select a set of words referring to objects, among which the pairwise shape similarity was orthogonal to the pairwise conceptual association.

Here, we differentiated two types of conceptual relations. One assumes concepts are componential, consisting of a set of shared semantic features (e.g., shape, action, motion, and emotion); similarity across semantic features leads to taxonomic relations or categories (e.g., forks and plates are manmade objects, not animals). The other assumes concepts are holistic; complementary roles within the same scenario lead to thematic relations (e.g., folks and plates relate to eating, not sleeping). This study focused on thematic relations by confining its stimuli to one taxonomic category—manmade objects, based on the following considerations: (1) Growing evidence suggests taxonomic and thematic relations rely on dissociable neural systems (e.g., [95,96,107]). Confusing two quality-different conceptual relations into one unified conceptual RDM might be problematic. (2) It is challenging for a neuroimaging study to distinguish whether a brain area represents taxonomic knowledge per se or shows preferences for specific taxonomic categories. Such distinction matters. The former assumes a dedicated brain area representing taxonomic relations among concepts, whereas the latter could mean a brain area representing category-specific features at the pre-conceptual stage. The category-specific

features could be the distinguishing features across brain systems. For example, as mainly manmade objects have manipulation-related features and humans have social features, brain systems processing action or social information will exhibit taxonomic preference. The category-specific features could also be within one brain system. For example, different patches along the visual pathway show category-specific effects (e.g., [108,109]) serving the perceptual purpose (e.g., [110]). However, in both cases, we can hardly say these brain systems represent taxonomic relations among concepts. Taxonomic representation at the conceptual level is assumed to emerge from the converge zones when multiple features have already been bound onto a concept (e.g., [95,111]). (3) Some hypothesis argues that category-specific representations at the conceptual level do exist; however, they cannot be represented in local brain areas but emerge from the connectivity among distributed categorical-specific regions across different brain systems [112]. This hypothesis explains why category-specific semantic deficits are well documented in neuropsychological literature (e.g., [113,114]) but are difficult to localize in the brain. However, category-specific representation at the conceptual level does not equal taxonomic knowledge representation, and the proposed connectivity-based neural representations are beyond the scope of this study. (4) Compared to the sighted, the early blind lack perceptual experience with many concepts in the natural world and have different neural representations of these "imperceptible" concepts [115]. Using only manmade objects ensures a relatively fair comparison between the sighted and the early blind (see ratings on touch experience in Fig 1B).

As a starting point, we preselected a set of Italian words referring to 60 everyday manmade objects based on our subjective impressions so that, among these objects, the shape similarity did not always correlate to the conceptual association. For example, a plate ("piatto") is perceptually similar to a coin ("moneta") but conceptually relates to a fork ("forchetta").

Next, we recruited 19 sighted native Italian speakers (age: M = 25.4, SD = 3.6) who did not participate in the fMRI experiments to rate the shape similarity and the conceptual association among the 60 objects. As pairwise rating among numerous items is time-consuming (60 objects require 1,770 pairs of comparison), we adopted the multi-arrangement method [116]. By doing so, participants arranged Italian words on a computer screen by mouse drag-and-drop operations in two 45-min task sessions. The closeness among the words was required to reflect shape similarity in the shape task session and conceptual association in the conceptual task session. Participants were instructed to disregard other object properties like color and size. The pairwise dissimilarity matrix of shape and conceptual information was estimated as the weighted mean of the scale-adjusted on-screen distances from individual arrangements. We averaged the ratings across participants and obtained a mean pairwise dissimilarity matrix for shape and conceptual information, respectively.

Then, these participants rated the potential confounding factors, i.e., object size (big versus small), toolness (tools versus non-tool manmade objects), and contextual association (strong versus weak contextual association objects). Participants were instructed to rate these three unidimensional variables by sliding a horizontal slider from left to right on a computer screen. To assess the variance in familiarity across objects, participants also rated each object on a 7-point Likert scale about the degree to which they knew its typical shape and primary function (1: do not know it at all; 7: know it very well). We also conducted a telephone interview with 16 early blind participants (8 females; age: M = 33.0, SD = 6.6; 6 of the participants took part in the fMRI experiment). We let them perform the same shape and conceptual familiarity rating tasks and asked them whether they had ever touched the objects. We averaged the rating score across participants to obtain a mean rating score for each object and each rating task.

After that, we selected 21 from the 60 Italian words based on the above ratings. This set of words met the following criteria: (1) Both sighted and early blind participants knew each

object's typical shape and primary function. The shape and the conceptual familiarity rating scores were higher than 5.8 (7-point Likert scale) in both groups. (2) Most early blind participants we interviewed (i.e., at least 14 among 16 participants) had touched the objects. (3) Shape similarity and conceptual association were orthogonal across pairs of objects. The absolute value of Spearman's correlation coefficient was 0.039. (4) Both shape similarity and conceptual association were orthogonal to the potential confounding factors, including both shape and conceptual familiarity from both early blind and sighted participants, word length (i.e., number of letters), word frequency (i.e., the Zipf value of the word occurrence in film and television subtitles; http://crr.ugent.be/subtlex-it/), object size, toolness, and contextual association. Since all these confounding factors were unidimensional, we measured the pairwise dissimilarity of these variables as the absolute difference between each pair of objects and correlated it to the shape and the conceptual information, respectively. The absolute values of Spearman's correlation coefficients were all below 0.15. (5) The variances across pairwise shape similarity (variance = 0.54) and pairwise conceptual association (variance = 0.53) were maximized while kept comparable. (6) Each object had at least one shape-matched item and one conceptual-associated item. S2 Table shows the 21 Italian words and their English translation.

Finally, a professional narrator recorded his pronunciation of these 21 words. We cut out the silence period at the beginning and the end of each auditory word with the same threshold and equalized the average intensity of all the auditory words as 70 dB using Praat 6.1.01 (https://www.fon.hum.uva.nl/praat/).

## Procedures

Before the fMRI scanning, all participants rated each object on a 7-point Likert scale about the degree to which they knew its typical shape and primary function (1: do not know it at all; 7: know it very well). They also rated how frequently they touched each object (1: have never touched it before; 7: touch it every day). We then explained the items of which either shape or conceptual familiarity rating score was below 6 points to ensure that all participants knew each object's typical shape and primary function. S1 Text shows the survey questions of these ratings.

During the fMRI scanning, we presented audio stimuli using Psychotoolbox-3 (http://psychtoolbox.org/). The sound was delivered through in-ear headphones. Before the formal scanning, we adjusted the volume for each participant so that they could hear the pronunciation clearly under the scanning noise but did not feel too loud. To ensure both sighted and blind participants received the same input during the scanning, we blindfolded all participants and turned off the lights in the scanning room.

The scanning session included one resting-state run at the beginning (8 min), 10 task-state runs (5 min 30 s each), and one run collecting T1 weighted images after the first five task-state runs (S1A Fig). During the resting-state run, participants were instructed to keep their heads still, not fall asleep, and not think about particular things. During the task-state runs, participants performed verification tasks on the words they heard.

Each task-state run was divided into two even blocks (S1A Fig). One corresponded to the shape verification task, and the other corresponded to the conceptual verification task. The order of the two task blocks was interleaved across runs within each subject, and whether the first run started with a shape or a conceptual block was counterbalanced across subjects within the EB and the SC group. Each block started with a 10 s rest, followed by a 20 s task probe. In the shape verification block, we instructed participants to think carefully about objects' shape ("Pensa attentamente alla forma") and judge whether they were elongated ("allungato"),

angular ("angolare"), hollow ("cavo"), circular ("circolare"), and discal ("discoidale"). In the conceptual verification block, we instructed participants to think carefully about objects' function ("Pensa attentamente alla funzione") and judge whether they were used for eating ("per mangiare"), writing ("per scrivere"), sleeping ("per dormire"), lighting ("per illuminazione"), and purchasing ("per fare acquisti"). These five shape and conceptual verification tasks were randomly assigned to each participant's first five task-state runs, and the second five task-state runs repeated these tasks in the same order. In this way, gaps between the same tasks were evenly distributed, and the same tasks could not be repeated in close time proximity. Participants made a yes/no judgment by pressing buttons using their right index/middle fingers. The button configuration (correspondence between yes/no judgments and index/middle fingers) in the first five runs was counterbalanced across subjects within the EB and the SC group. To counterbalance the motor effects of different fingers within subjects, we told each participant that the button configuration was switched in the second five runs.

Each block included 21 trials after the task probe, with 21 words presented once (S1B Fig). Each trial started with a 100 ms beep to capture participants' attention, followed by a 300 ms silence and an auditory word (word duration: M = 662 ms, SD = 165 ms). The stimulus onset asynchrony was jittered as either 5 s or 8 s—11 trials lasted 5 s, and 10 trials lasted 8 s. The order of the words and the jitter intervals were randomized for each block. Participants were instructed to press buttons within 5 s after the stimulus onset. The RT was measured as the interval between the stimulus onset and the button press.

After the fMRI scanning, participants also rated object properties. For pairwise shape similarity and pairwise conceptual association, we adapted the paradigm for both sighted and blind populations by presenting the stimuli in the auditory modality. In each trial, participants heard two words in sequence and rated on a 7-point Likert scale (for shape rating, 1: not similar at all, 7: identical in shape; for conceptual rating, 1: not associated at all, 7: strongly associated). Both rating tasks consisted of 210 trials covering all the possible object pairs. For the other three object properties as potential confounding factors, participants rated item-wise on a 7-point Likert scale. They were object size (1: as small as a needle, 7: as big as a television), toolness (1: non-tools like a lamp, 7: tools like a hammer), and conceptual association (1: weak contextually associated like a cellphone, 7: strong contextually associated like a bowling ball). S1 Text shows the survey questions of these ratings.

## Behavior analysis

For pairwise shape similarity and conceptual association ratings, we averaged the rating scores across all participants who took part in the fMRI experiment and calculated the model RDMs for the following RSA (i.e., 7 minus the mean rating score). To investigate the organizational structure of the two model RDMs, we performed the clustering analysis using the k-means clustering algorithm [38,39]. The maximum number of iterations was 10,000, the number of times to repeat clustering using new initial cluster centroid positions was 100, and the silhouette criterion was adopted to decide the optimal number of clusters in the range from 2 to 10 [40]. We conducted this analysis using the *kmeans* and *evalclusters* function in Matlab 2021.

For the ratings on other object properties (i.e., object size, contextual association, and toolness) and touch experience, we averaged the rating scores across all participants to obtain a mean vector for each rating item. These mean rating vectors, together with word duration and word frequency, constituted the potential confounding factors. To investigate the effect of these factors in the subsequent parametric modulation analysis, we orthogonalized these unidimensional variables using principal component analysis. Varimax rotation was applied to increase the interpretability of components, and five RCs of which the eigenvalues were greater

than 1 were selected. The principal component analysis was performed using the *principal* function in the R package *psych 2.1.9*.

Analyses of the performance during fMRI scanning were conducted using JASP (Version 0.16).

## MRI acquisition

MRI data were acquired using a MAGNETOM Prisma 3T MR scanner (Siemens) with a 64-channel head-neck coil at the Center for Mind/Brain Sciences, University of Trento. Functional images were acquired using the simultaneous multislices echoplanar imaging sequence: the scanning plane was parallel to the bicommissural plane, the phase encoding direction was from anterior to posterior, repetition time (TR) = 1000 ms, echo time (TE) = 28 ms, flip angle (FA) = 59˚, multiband factor = 5. All participants in the early blind and sighted control groups and seven participants in the independent sighted group used a 3 mm spatial resolution: field of view (FOV) = 198 mm × 198 mm, matrix size = 66 × 66, 65 axial slices, slices thickness (ST) = 3 mm, gap = 0.3 mm, voxel size = 3 × 3 × (3 + 0.3) mm. The rest nine participants in the independent sighted group used a 2 mm spatial resolution: FOV = 200 mm × 200 mm, matrix size = 100 × 100, 65 axial slices, ST = 2 mm, gap = 0.2 mm, voxel size = 2 × 2 × (2 + 0.2) mm. Three-dimensional T1-weighted images were acquired using the magnetization-prepared rapid gradient-echo sequence, sagittal plane, TR = 2,140 ms, TE = 2.9 ms, inversion time = 950 ms, FA = 12˚, FOV = 288 mm × 288 mm, matrix size = 288 × 288, 208 continuous sagittal slices, ST = 1 mm, voxel size = 1 × 1 × 1 mm.

## MRI preprocessing

We performed MRI preprocessing using fMRIPrep 20.0.5 ([117]; RRID: SCR_016216), based on Nipype 1.4.2 ([118]; RRID: SCR_002502). Please see S2 Text, a boilerplate text directly generated by the fMRIPrep. It describes the detailed preprocessing steps used in the current study, aiming for a clear and consistent description to improve experimental reproducibility.

As surface-based analysis can significantly improve the spatial localization compared to the traditional volume-based analysis [119], we analyzed the images in the surface space generated by fMRIPrep (i.e., the fsaverage5 or the fsnative space). We conducted the surface smoothing of the functional images with a full width at half maximum (FWHM) of 6 mm using the mri_surf2surf command in FreeSurfer (http://surfer.nmr.mgh.harvard.edu/).

## First-level neuroimaging analysis

We performed the first-level analysis using SPM12 (https://www.fil.ion.ucl.ac.uk/spm/software/spm12/). Individual-level GLMs were built separately for univariate contrast, parametric modulation, and RSA. In all three GLMs, six rigid-body transformation parameters and constant variables indicating each of the 10 runs were involved as nuisance regressors. A high-pass filter with a cutoff of 512 s was used to remove low-frequency noise and slow drifts. The RSA used unsmoothed images, while the other two analyses used smoothed images.

The GLM for the univariate contrast analysis involved three events—the shape task, the conceptual task, and the task probe. The duration of shape and conceptual tasks was set as each trial's RT, and the duration of task probes was set as the auditory period before each block introducing the task ahead. The resulting boxcar function was convolved with a canonical hemodynamic response function (HRF). In this way (i.e., the variable epoch approach), the trial-by-trial RT variability was modeled [46]. To further control the domain-general effect of RT across the two tasks, we also used stick functions to model the trial-by-trial RT variability. We pooled the trials in the two tasks together, modulated the amplitude of sticks by the mean-

centered RT, and convoluted the RT-modulated stick function with the canonical HRF (i.e., the variable impulse approach). The resulting RT variable was involved in the GLM as one regressor. We contrasted the shape task, the conceptual task, and the RT regressor to the resting state and contrasted between shape and conceptual tasks. The obtained combined beta images were used in the second-level analysis.

The GLM for the parametric modulation analysis only involved two conditions—the trials (i.e., shape and conceptual tasks pooled together as one condition) and the task probes. The duration of trials was set as its RT, and the duration of task probes was set as the auditory period before each block introducing the task ahead. We modulated the condition of the trials with a set of parametric variates, including the task type (i.e., the shape task coded as one and the conceptual task coded as −1), the z-scores of the RT across all the trials in each run, the RCs corresponding to word duration, word frequency, object size, toolness, and touch experience. The option for orthogonalizing modulations in the SPM was turned off [120]. We contrasted each parametric modulator to zero. The obtained combined beta images were used in the second-level analysis.

The GLM for the RSA involved each word in each task as a separate condition and the task probes as one condition. We concatenated 10 runs to improve the reliability of the model estimation [35]. The duration of trials was set as its RT, and the duration of task probes was set as the auditory period before each block introducing the task ahead. The trial-by-trial RT variability across the two tasks was also modeled using the variable impulse approach. We contrasted each word in each task to the resting state. The obtained T images instead of the beta images were used in the following RSA [121].

## Representational similarity analysis

The RSA was conducted among the 21 object conditions within shape and conceptual tasks separately. It included two steps of correlation [35]. In the first-order correlation, we calculated the Spearman distance of the activity patterns across vertices between each pair of conditions and obtained a 21 × 21 neural RDM for a particular region. In the second-order correlation, we correlated the neural RDM and each model RDM (i.e., shape similarity and conceptual association) across the 210 pairs using Spearman correlation. The resulting correlation coefficients were Fisher z-transformed using the inverse hyperbolic function.

The ROI-based RSA focused on two sets of ROIs derived from significant brain areas in the second level of the parametric modulation analysis (see below). The shape ROIs were bilateral and had significantly greater activation in the shape task than in the conceptual task—the ILOTC, the aIPS, the pIPS, and the vPMC. The conceptual ROIs were left-lateralized and were significant in the opposite contrast—the orbital IFG, the aLTC, the pSTG, the AG, and the SMG. In cases when clusters were stuck together under the conventional threshold (vertex-wise $p < 0.001$, cluster-level FWE corrected $p < 0.05$), we raised the vertex-wise threshold until they were isolated.

The searchlight-based RSA was performed to provide a global view of the results [122]. The searchlight spot went through all the vertices on the fsaverage5 surface. For each vertex, the spot included the six vertices directly connecting to the central vertex and the more peripheral vertices connecting to the six vertices (i.e., 19 vertices in total) [123]. The Fisher z-transformed second-order correlation coefficient was assigned back to the central vertex, and a surface smoothing with a 6 mm FWHM was applied to the resulting maps.

To investigate whether the ILOTC in EB and SC represented the same content, we compared the inter-subject neural RDM correlation within the same group (i.e., EB-EB and SC-SC) and between different groups (i.e., EB-SC). The within-group inter-subject correlation

was calculated in a leave-one-subject-out manner. The neural RDM of the ILOTC of one participant was correlated to the mean neural RDMs of all the other participants within the same group across the 210 object pairs. This procedure ended up with 16 correlation coefficients for each group. The between-group inter-subject correlation was calculated in two steps. First, the neural RDM of the ILOTC of each participant in one group was correlated to the mean neural RDMs of all the participants in the other group across the 210 object pairs, which generated 16 correlation coefficients for each group. Second, we averaged the correlation coefficients from the EB and SC participants in the same pair to obtain 16 between-group correlation coefficients. These correlation coefficients were calculated using Spearman's correlation and were Fisher z-transformed.

To provide a planar visualization of the representational pattern in bilateral ILOTC in the shape task, we performed the multidimensional scaling analysis using the *mdscale* function in Matlab 2021. The input dissimilarity matrix was the mean Euclidean distance between each pair of conditions averaged across all participants ($N = 48$). We used the squared stress, normalized with the sum of fourth powers of the dissimilarities, as the goodness-of-fit criterion to minimize.

## Resting-state functional connectivity

We started with the unsmoothed resting-state images. To remove nonneuronal nuisance variables, we built a GLM to predict the timecourse of each vertex using the 24 head motion regressors [124], the mean timecourses in a conservative mask of the white matter and the cerebrospinal fluid extracted by the fMRIPrep, and the linear trend with the time points. We estimated the beta coefficients using the *fitglm* function in Matlab 2021 and subtracted all the terms (i.e., the dot product of all the nuisance variables and their estimated beta coefficients) from the original timecourses. A band-pass filter (0.01 to 0.1 Hz) was then performed on the resulting timecourses using the infinite impulse response filter method, and surface smoothing was carried out with a 6 mm FWHM. The functional connectivity between the two regions was defined as Pearson's correlation between their timecourses. The correlation coefficients were Fisher z-transformed before the second-level analysis. The ROIs used in the seed-based RSFC and the interregional RSFC also came from the parametric modulation analysis.

## Second-level neuroimaging analysis

We performed the group-level one-sample test or two-sample test (i.e., EB versus SC) on the first-level beta images from the univariate contrast and parametric modulation analyses, the searchlight-based RSA images, and the RSFC images. The statistic inference was made using the permutation method with PALM (https://fsl.fmrib.ox.ac.uk/fsl/fslwiki/PALM). Five thousand sign flips were performed [125]. It is worth noting that, in the two-sample tests, we also chose the sign-flipping method assuming independent and symmetric errors instead of the traditional permutation method assuming exchangeable errors. This is because the variance of the early-blind group, on many occasions, is greater than the variance of the sighted-control group (Fig 4C; e.g., [126]), which violates the equal variance assumption of exchangeability. For the $p$-value below 0.01, we fit a generalized Pareto distribution to model the tail of the permutation distribution, aiming to improve the precision of the $p$-values [127,128].

In most cases, we controlled the FWE rate using a conventional cluster-forming threshold (i.e., vertex-wise $p < 0.001$, cluster-level FWE corrected $p < 0.05$). In the cases when the cluster-forming threshold was not suitable (i.e., distributed clusters spliced together), we controlled the FWE rate using a more conservative vertex-wise threshold (i.e., vertex-wise FWE corrected $p < 0.005$). We corrected the multiple comparisons of the two hemispheres using

Bonferroni correction—the threshold set on each hemisphere was vertex-wise $p < 0.001$, cluster-level FWE corrected $p < 0.025$, or vertex-wise FWE corrected $p < 0.0025$.

## Brain visualization

The brain results were illustrated using the Connectome Workbench 1.5.0 (https://www.humanconnectome.org/software/connectome-workbench). We mapped the significant brain areas from the fsaverage5 surface to the fsLR surface using the ADAP_BARY_AREA method for visualization purposes. They were displayed on an inflated surface against the group-averaged all sulcus image of 1,096 young adults from the dataset of the Human Connectome Project (https://balsa.wustl.edu/reference/pkXDZ).

## Supporting information

**S1 Fig. The procedure of the fMRI experiment.** (A) The structure of the fMRI scanning session. The order of the two task blocks was interleaved across runs within each subject. Whether the first run started with a shape or a conceptual block was counterbalanced across subjects within the early blind and the sighted control group. Task probes S1 to S5 randomly corresponded to the five questions about objects' shape for each participant (i.e., is the object elongated, angular, hollow, circular, and disc-shaped?). Task probes C1 to C5 randomly corresponded to the five questions about objects' function for each participant (i.e., is the object used for eating, writing, sleeping, lighting, and purchasing?). The button configuration (correspondence between yes/no judgments and index/middle fingers) in the first five runs was counterbalanced across subjects within the early blind and the sighted control group. The button configuration was switched in the second set of five runs (after T1 acquisition) for each participant. (B) The timing of each block and each trial. The participants were instructed to respond by pressing buttons within 5 s. In this figure, the speaker icon is from Wikimedia Commons (https://commons.wikimedia.org/wiki/File:Speaker_Icon.svg) and the button-press icon is from Flaticon (https://www.flaticon.com/free-icon/press-with-two-fingers_4622).
(PDF)

**S2 Fig. The vPMC-IPS circuit showed greater activation in the shape task than in the conceptual task in the early blind (EB).** (A) ROIs defined in the contrast between shape and conceptual tasks in the sighted control (SC) (vertex-wise $p < 0.001$, cluster-level FWE corrected $p < 0.05$). (B) ROI analyses in the contrast between shape and conceptual tasks in EB using the ROIs defined in SC. *: $p < 0.05$, **: $p < 0.01$. The underlying data for this figure can be found in S1 Data.
(PDF)

**S3 Fig. Early blind (EB) versus sighted control (SC) in shape and conceptual tasks (vertex-wise $p < 0.001$, cluster-level FWE corrected $p < 0.05$).** (A) EB versus SC in the shape task. (B) EB versus SC in the conceptual task. The underlying data for this figure can be found in S1 Data.
(PDF)

**S4 Fig. Neural correlates of reaction time (RT, vertex-wise $p < 0.001$, cluster-level FWE corrected $p < 0.05$).** (A) The RT effect across all the participants ($N = 48$). (B) The differences in the RT effect between the early blind (EB) and the sighted control (SC). The underlying data for this figure can be found in S1 Data.
(PDF)

**S5 Fig. Overlap between the conceptual network and the language network.** The conceptual task in our study involved a brain network (in blue) almost identical to the language network (in purple), except for the triangular part of the IFG and the 55b area in the premotor cortex. These two dorsal regions are considered to play a non-semantic role in language processing. The conceptual network (in blue) was defined in the contrast between the conceptual task and the shape task with the control of other object properties ($N = 48$; vertex-wise $p < 0.001$, cluster-level FWE corrected $p < 0.05$). The language network (in purple) was defined in the study by Fedorenko and colleagues [48] with the data updated from 220 participants. The overlap coefficient between these two networks was 83.05%. Such highly overlapped results suggest that the language system plays a crucial role in our conceptual task.
(PDF)

**S6 Fig. Neural correlates of linguistic variables (vertex-wise $p < 0.001$, cluster-level FWE corrected $p < 0.05$).** (A) Neural correlates of word duration. (B) Neural correlates of word frequency. The underlying data for this figure can be found in S1 Data.
(PDF)

**S7 Fig. RSA results of shape similarity in the brain areas with greater activation in the shape task than in the conceptual task across all participants ($N = 48$).** (A) Brain areas with significantly greater activation in the shape task than in the conceptual task defined in Fig 2A. (B) RSA results of these shape-relevant areas in shape and conceptual tasks. *: $p < 0.05$, **: $p < 0.01$, ***: $p < = 0.001$. The underlying data for this figure can be found in S1 Data.
(PDF)

**S8 Fig. Whole-brain searchlight results of shape similarity.** (A) Whole-brain searchlight results of shape similarity across all participants ($N = 48$; vertex-wise FWE corrected $p < 0.005$, cluster size $> 400$ mm$^2$). (B) Group difference of whole-brain searchlight of shape similarity between the early blind (EB) and the sighted control (SC) (vertex-wise $p < 0.001$, cluster-level FWE corrected $p < 0.05$). The underlying data for this figure can be found in S1 Data.
(PDF)

**S9 Fig. Whole-brain searchlight results of conceptual association across all participants ($N = 48$; vertex-wise $p < 0.001$, cluster-level FWE corrected $p < 0.05$).** The underlying data for this figure can be found in S1 Data.
(PDF)

**S10 Fig. Contrast between the early blind (EB) and the sighted control (SC) in the seed-based RSFC results from the left ILOTC (vertex-wise $p < 0.001$, cluster-level FWE corrected $p < 0.05$).** The underlying data for this figure can be found in S1 Data.
(PDF)

**S1 Table. Demographic information of the early blind and their matched sighted control.**
(PDF)

**S2 Table. Stimuli.**
(PDF)

**S3 Table. Inter-rater reliability of object properties and touch experience within each group of participants.**
(PDF)

**S4 Table. Neural representation in bilateral aIPS.**
(PDF)

**S5 Table. Neural representation in bilateral pIPS.**
(PDF)

**S6 Table. Neural representation in bilateral vPMC.**
(PDF)

**S1 Text. English translation of survey questions.**
(PDF)

**S2 Text. MRI preprocessing using fMRIPrep.**
(PDF)

**S1 Data. Source data underlying the figures.**
(ZIP)

## Acknowledgments

We are thankful to our blind participants. We also thank the Unioni Ciechi of Trento, Mantova, Genova, Savona, Cuneo, Torino, Trieste, Milano, and the Institute of the Blind in Milan for helping with the recruitment. We are grateful to Jorge Jovicich for technical assistance in developing fMRI acquisition sequences, Martina Zanotto and Anna D'Urso for their help in data collection, and Edoardo Camponeschi for recording the stimuli.

## Author Contributions

**Conceptualization:** Yangwen Xu, Roberto Bottini, Olivier Collignon.

**Data curation:** Yangwen Xu, Federica Sigismondi.

**Formal analysis:** Yangwen Xu, Lorenzo Vignali.

**Funding acquisition:** Davide Crepaldi, Olivier Collignon.

**Investigation:** Yangwen Xu, Lorenzo Vignali, Davide Crepaldi, Roberto Bottini, Olivier Collignon.

**Project administration:** Roberto Bottini, Olivier Collignon.

**Resources:** Roberto Bottini.

**Supervision:** Davide Crepaldi, Roberto Bottini, Olivier Collignon.

**Validation:** Lorenzo Vignali.

**Visualization:** Yangwen Xu.

**Writing – original draft:** Yangwen Xu, Olivier Collignon.

**Writing – review & editing:** Yangwen Xu, Lorenzo Vignali, Federica Sigismondi, Davide Crepaldi, Roberto Bottini, Olivier Collignon.

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
