## [Editor Report · Decision Letter 0]

18 Nov 2022

Dear Dr Xu, 

Thank you for submitting your manuscript entitled "Supramodal Shape Representation in the Human Brain" for consideration as a Research Article by PLOS Biology.

Your manuscript has now been evaluated by the PLOS Biology editorial staff, as well as by an academic editor with relevant expertise, and I am writing to let you know that we would like to send your submission out for external peer review.

Once your full submission is complete, your paper will undergo a series of checks in preparation for peer review. After your manuscript has passed the checks it will be sent out for review. To provide the metadata for your submission, please Login to Editorial Manager (https://www.editorialmanager.com/pbiology) within two working days, i.e. by Nov 20 2022 11:59PM.

Kind regards,

Kris

Kris Dickson, Ph.D., (she/her)

Neurosciences Senior Editor/Section Manager

PLOS Biology

kdickson@plos.org

---

## [Decision Letter · Decision Letter 1]

17 Jan 2023

Dear Dr Xu,

Thank you for your patience while your manuscript "Supramodal Shape Representation in the Human Brain" was peer-reviewed at PLOS Biology. It has now been evaluated by the PLOS Biology editors, an Academic Editor with relevant expertise, and by several independent reviewers. 

In light of the reviews, which you will find at the end of this email, we would like to invite you to revise the work to thoroughly address the reviewers' reports. Given the extent of revision needed, we cannot make a decision about publication until we have seen the revised manuscript and your response to the reviewers' comments. Your revised manuscript is likely to be sent for further evaluation by all or a subset of the reviewers.

**IMPORTANT - SUBMITTING YOUR REVISION**

*Re-submission Checklist*

*Published Peer Review*

*PLOS Data Policy*

*Blot and Gel Data Policy*

Sincerely,

Kris

Kris Dickson, Ph.D., (she/her)

Neurosciences Senior Editor/Section Manager

PLOS Biology

kdickson@plos.org

REVIEWS:

Reviewer's Responses to Questions

Do you want your identity to be public for this peer review?

Reviewer #1: No

Reviewer #2: No

Reviewer #3: No

Reviewer #1: The authors perform an fMRI study on blind and sighted participants to whom they present spoken words in the scanner that participants are asked to rate according to their shape similarity and contextual similarity. The authors find stronger responsiveness of the visual ILOTC region during the shape compared to the contextual task both in sighted and blind individuals. This study is interesting and advances the field, but needs revision with respect to the way the authors conceptualize their findings.

Major concerns

Concept. The concept that is here associated to the conducted analysis and obtained results is a bit misleading. The term "supramodal shape representation" suggests that shape representations are investigated that occur across modalities - however, the stimuli were only auditory. Given the authors start with saying that objects can be perceived by either vision or touch, and they do not mention in the abstract that in this study, stimuli were only auditory, the reader is led to believe that shape representation was tested in multiple modalities. This is similar to the first paragraph of the introduction, where the question is posed whether the brain can represent object information independent from the senses - this cannot be investigated testing on one sensory modality only. To test for supramodal representations of objects, one would imagine a task where specific objects are either haptically explored or heard, which would then assume to activate similar brain areas. Here, what the authors really investigate is to imagine the shape of an object or to imagine the context in which the object is used based on hearing the name of an object. Given that the shape task is a clear visual object feature task (elongated, angular, hollow, circular, discal) and the conceptual task is clearly related to object use (eating, writing, sleeping), it is not surprising that a visual area is activated more in the former condition. This, however, cannot be interpreted as a supramodal representation of objects. I would therefore suggest to adapt the concept and title to highlight that the ILOTC is involved in shape representation both in blind and sighted individuals. This also fits to the correlation to object size perception reported. How "supramodal" this representation is has to be clarified by other studies.

In a similar vein, the authors pose the hypothesis in the introduction that potentially, the ILOTC may be more responsible for conceptual representations in blind compared to sighted individuals. This is still possible as one could imagine that the representation of the object's shape induced by haptic object exploration is more pronounced in ILOTC in blind compared to sighted individuals. This could only be tested by an experiment that presents the same object in different modalities, as pointed out above. This hypothesis can therefore also not be clarified in this study, which requires re-writing the introduction in parts.

It is also not clear to me why the authors refer to the conceptual task as "linguistic". When hearing the word of an object, and being asked whether this object is used for eating, sleeping, writing etc, this is not a semantic task but it is really about imaging the use of the object in everyday life. The interpretation of this being a "semantic" task is therefore highly speculative. This can be mentioned in a balanced way in the discussion, but should for sure not be reported as a result in the results section (line 278 and subsequent, see also line 319 where the authors also give a strong and rather questionable interpretation of the conceptual task being "higher-level cognitive"). This needs to be changed in the results and discussion sections.

Results. The interaction between group and task was reported as non-significant with a p-value of 0.123. Given the groups were relatively small, there may be an interaction effect that is not fully revealed here based on undersampling. It would therefore be good if the authors would report the group size that would be needed to find a significant interaction effect, and if they would perform a Bayesian analyses showing that indeed the absence of an effect has a higher likelihood that the presence of an effect. In addition, it should be reported in the text where this trend towards an interaction comes from.

The same is true for the comparison between SC-SC and EB-SC, where a p-value of 0.073 is interpreted as "no effect". Please apply the same methods as suggested above here, and adapt the interpretation based on the results of these analyses.

Discussion. Based on the comments and suggestions mentioned above, the discussion needs to be adjusted.

Minor comments

In Figure captions, the short forms EB and SC are not explained 

Reviewer #2: In this work, Xu and colleagues investigate whether the modality-independent recruitment of the inferior lateral occipitotemporal cortex (ILOTC) relates to the processing of shape features rather than to the conceptual representation of objects. To address this question, the authors ask early blind and sighted individuals to perform a shape categorization task (e.g., round vs. square shape) and a conceptual verification task (e.g., it is for eating vs. writing) on 21 manmade objects while their brain activity is recorded using fMRI. Shape and conceptual similarity judgments collected in behavioral experiments complement the imaging experiment, and brain data are analyzed using univariate and multivariate (i.e., RSA) methods.

Results demonstrate (1) that ILOTC is recruited to a greater extent when participants focus on shape features, (2) that the response pattern in this region relates to the similarity in shape more than to the similarity in conceptual features, and (3) that this area is specifically connected at rest with a network of shape-relevant frontal and parietal areas. Importantly, these results are observed in sighted and early blind individuals.

Overall, this is a very well-conducted study and a relevant contribution to the literature on the brain representation of object features. The study hypothesis is clearly explained, the experimental paradigm is well-conceived, and the analyses are sound. The results are interesting, and their interpretation is convincing.

Here, I am providing a list of comments that the authors may want to incorporate in a revised version of the manuscript.

1. I believe the paragraph about parietal regions (lines 56-69) in the "Introduction" section could be moved to the discussion section. Because the focus of this investigation is ILOTC (e.g., LOtv peak coordinates extracted from previous studies are used as a reference), this paragraph sounds like an unnecessary digression from the main topic.

2. The paragraph detailing the fMRI paradigm in the "Materials and Methods" section is not sufficiently clear to me (lines 765-789). I feel adding a figure that recapitulates the fMRI paradigm would help in clarifying the structure (e.g., randomization of task conditions) and timing of the different experimental phases (e.g., probe, stimulus presentation, participant response).

3. Concerning the assessment of statistical significance, the authors write: "We performed the group-level one-sample t-test or two-sample t-test[...]. Five thousand sign-flips were performed[...]" (lines 958-964). To my knowledge, sign-flipping is used only in the case of one-sample or two-sample paired non-parametric t-test (after having computed the difference between time points for each participant). Thus, it is not clear what Xu and colleagues mean when they refer to sign flips in the context of a two-sample unpaired t-test. Do they mean the shuffling of labels?

4. Although the results on the modality- and experience-independent representation of shape in ILOTC are convincing and in line with previous findings, the fact that the authors have used spoken words to trigger mental representations of objects limits the possibility of understanding which shape features are actually encoded in ILOTC. For instance, even though silhouette, curvature, and medial axis often correlate with each other, they may contribute differently to the final percept and could be mapped in distinct brain areas (see Papale et al., 2020; DOI: 10.1152/jn.00212.2020). In particular, medial-axis - a property that well accounts for behavioral similarity judgments and transformation-resistant shape descriptors - seems to be encoded in LO. I believe the discussion section would benefit from a more thorough analysis of the potential impact that having used memory-related activations of mental representations of objects, rather than actual perception (e.g., tactile match to sample vs. tactile recognition tasks), may have had on the present findings.

Reviewer #3: The report by Xu et al., "Supramodal Shape Representation in the Human Brain," presents results from a study of the representation of manipulable and graspable manmade objects in the brains of early blind and sighted control participants that investigates whether the supramodal representation in the the lateral occipital tactile visual complex (LOtv), and more broadly the inferior lateral occipital temporal complex (ILOTC) reflects information about shape, conceptual knowledge, or both. The study analyzed neural responses, as measured with fMRI, to auditory words for 21 objects, during tasks that focused processing on shape or conceptual attributes. Results from both univariate and RSA analysis show strong evidence for shape representation that is independent of conceptual features, and the authors conclude that ILOTC represented shape similarity, not conceptual association, in both the early blind and sighted control participants

This is a well-designed and comprehensive study, and the conclusion that the representation in ILOTC reflects supramodal shape information is well-supported. The other conclusion, however, that "reject[s] the alternative hypotheses that such activation depends on visual imagery or conceptual processing," is too strong. "Conceptual processing" here is operationalized in a very narrow domain, namely concepts concerning the typical settings in which manipulable and graspable manmade objects are encountered. It is well-known that conceptual information is encoded in LOC cortex. The domains of the conceptual knowledge that have been clearly demonstrated, however, concern entities that are not relevant to tactile-visual representation, namely animal taxonomy, animacy, agency, and behavior (Kiani et al. 2007; Connolly et al. 2012; Sha et al. 2015; Nastase et al. 2017; Thorat et al. 2019). These entities and concepts are typically not experienced through touch and their representation in the early blind may be, as of now, largely unexplored. The full range of conceptual knowledge about manmade objects also is not well-sampled by their task, which concerns only the contexts in which these objects are encountered, but not how they are manipulated, what is their function, what are they made of, what is their typical color, etc. The neural representational geometry of conceptual associations among manmade objects is not well understood. In Kiani et al.'s seminal study in monkeys, they found a highly meaningful conceptually-driven geometry in monkey IT cortex for animals but nothing for objects, which could be attributed to monkey's limited knowledge about manmade objects. The results in the current study show that the representational geometry of conceptual associations is mostly negative except, curiously, for areas in the default mode system (TPJ, precuneus, MPFC) that were also involved in representation of shape knowledge. Consequently, the conclusion of this study cannot address the representation of conceptual information in LOC more broadly. Their conclusion supporting a negative hypothesis, therefore, should be more carefully focused on the limited conceptual domain that they studied and not generalized to representation of conceptual knowledge in LOC. 

Minor points.

Line . The role for the 16 independent sighted (IS) participants is unclear. They are included in results for all 48 participants, but results restricted to this group are not presented.

Line 492. "similarly" should probably be "similarity"

Figure 4D. Why are there two MDS plots? Are they for right and left ILOTC or for a 3-dimensional MDS?

---

## [Decision Letter · Decision Letter 2]

2 Jun 2023

Dear Dr Xu,

Thank you for your patience while we considered your revised manuscript "Shared Neural Representations of Object Shape between the Sighted and Early Blind" for publication as a Research Article at PLOS Biology. This revised version of your manuscript has been evaluated by the PLOS Biology editors, the Academic Editor and two of the original reviewers.

The reviewers and our Academic Editor are fully satisfied by the revision and suggest we accept the manuscript. However, before we can editorially accept your study, we need you to address a few remaining data and other policy-related requests, which I outline below, in another revision that we think will not take very long.

**Please address the following editorial requests:

1) FINANCIAL DISCLOSURE: Please update your financial disclosures statement, in our online system, to describe the role of any sponsors or funders in the study design, data collection and analysis, decision to publish, or preparation of the manuscript. If the funders had no role in any of the above, include this sentence at the end of your statement: "The funders had no role in study design, data collection and analysis, decision to publish, or preparation of the manuscript."

2) ETHICS STATEMENT: Please update the ethics statement, contained in your methods section, to indicate whether the study was conducted according to the principles expressed in the Declaration of Helsinki.

3) BLURB: When resubmitting, in our online system, please provide a blurb which (if accepted) will be included in our weekly and monthly Electronic Table of Contents, sent out to readers of PLOS Biology, and may be used to promote your article in social media. The blurb should be about 30-40 words long and is subject to editorial changes. It should, without exaggeration, entice people to read your manuscript. It should not be redundant with the title and should not contain acronyms or abbreviations.

4) DATA AVAILABILITY: Thank you for providing the data underlying your figures as a series of supplemental tables. I noticed that the data provided for Fig 1 does not have labels on the columns/rows. Can you please update this?

We expect to receive your revised manuscript within two weeks. 

*Published Peer Review History*

*Press*

Sincerely,

Luke

Lucas Smith, Ph.D.

Senior Editor,

lsmith@plos.org,

PLOS Biology

Reviewer remarks:

Reviewer #1: Thanks a lot, the authors have addressed all of my comments.

Reviewer #2, Luca Cecchetti: I thank the authors for having addressed all my concerns.

I am now even more convinced about the correctness of the analysis pipeline and of the substantial contribution of this work to the understanding of object shape representation in the brain.

---

## [Editor Report · Decision Letter 3]

23 Jun 2023

Dear Dr Xu,

Thank you for the submission of your revised Research Article "Similar object shape representation encoded in the inferolateral occipitotemporal cortex of sighted and early blind people" for publication in PLOS Biology and thank you for addressing our previous editorial requests in this revision. On behalf of my colleagues and the Academic Editor, Frank Tong, I am pleased to say that we can in principle accept your manuscript for publication, provided you address any remaining formatting and reporting issues. These will be detailed in an email you should receive within 2-3 business days from our colleagues in the journal operations team; no action is required from you until then. Please note that we will not be able to formally accept your manuscript and schedule it for publication until you have completed any requested changes.

**As one last minor editorial request, which I forgot to include in my last email - please update each figure legend (including supplemental) to indicate where the underlying data can be found. For example, you can add the sentence "the underlying data for this figure can be found in S1_Data"

PRESS

Sincerely, 

Lucas Smith, Ph.D.

Senior Editor

PLOS Biology

lsmith@plos.org